# A Simplified, Efficient Approach to Hybrid Wind and Solar Plant Site Optimization

Charles Tripp[1], Darice Guittet[1], Jennifer King[1], and Aaron Barker[1]

[1]National Renewable Energy Laboratory (NREL), 15013 Denver West Parkway , Golden, CO 80401, United States of America

**Correspondence:** Charles Tripp (charles.tripp@nrel.gov)

**Abstract.** Wind plant layout optimization is a difficult, complex problem with a large number of variables and many local minima. Layout optimization only becomes more difficult with the addition of solar generation. In this paper, we propose a parameterized approach to wind and solar hybrid power plant layout optimization that greatly reduces problem dimensionality while guaranteeing that the generated layouts have a desirable regular structure. Thus far, hybrid power plant optimization research has focused on system sizing. We go beyond sizing and present a practical approach to optimizing the physical layout of a wind-solar hybrid power plant. We argue that the evolution strategies class of derivative-free optimization methods is well-suited to the parameterized hybrid layout problem, and we demonstrate how hard layout constraints (e.g. placement restrictions) can be transformed into soft constraints that are amenable to optimization using evolution strategies. Next, we present experimental results on four test sites, demonstrating the viability, reliability, and effectiveness of the parameterized evolution strategies approach for generating optimized hybrid plant layouts. Completing the tool kit for parameterized layout generation, we include a brief tutorial describing how the parameterized evolutionary approach can be inspected, understood, and debugged when applied to hybrid plant layouts.

## 1 Introduction

Hybrid power plants (HPPs) utilize multiple electrical generation methods to take advantage of each method's benefits while mitigating drawbacks of each individual method. Deployment of integrated hybrid renewable energy systems (HRES) is expected to increase because of their potential to improve flexibility, resilience, and economics. The diversity of generation resource, the potential for complementary overbuild and the importance of forecasting and control of HPP may provide value stacking and risk mitigation for the plant owner, as well as increased dispatch efficiency for the bulk grid. While solar photovoltaic (PV) with battery storage is the most common type of HPP, an increasingly prevalent hybrid combination is the combination of wind and solar. Wind-solar hybrid plants benefit from resource complementarity as well as shared permitting, siting, equipment, interconnection, transmission, and transaction costs. However, it can be difficult to optimally site wind-solar plants due to their higher combined complexity (Gorman et al., 2020). The design considerations of the stand-alone wind and solar plant apply to the hybrid plant in addition to those imposed by their colocation, such as sizing and the effect of wind turbine shading on solar energy performance. The turbines' layout, wind conditions, and operations are key to the wind plant's

annual energy production (AEP). Losses due to wake effects are a major factor, with (Clifton et al., 2016) showing it to be the loss factor with the highest maximum value and highest variation. Photovoltaic array design and effective irradiance are important site considerations for solar plants. Irradiance reduction can be estimated by on-site surveys or 3D models and minimized by reducing the ground coverage ratio (GCR), using tracking, and by reducing internal and external shading. Mismatch losses due to different electrical properties in shaded and unshaded portions can be estimated with module current-voltage models and minimized by module design and power electronics (MacAlpine et al., 2013; Bendib et al., 2015; Hanson et al., 2014). Matching components and balance-of-plant equipment with expected operational conditions, such as aging and resource availability, is an important step in HRES design that is not undertaken by this work but for which the approach and tools provided here could be adapted.

Optimizing a single technology alone is a challenging task. In particular, wind plant layout optimization has been addressed in recent literature to maximize the power output, minimize levelized cost of energy, or maximize expected profit Herbert-Acero et al. (2014); Chen and MacDonald (2014); Padrón et al. (2019); Nagpal et al. (2021); Croonenbroeck and Hennecke (2021). The wind plant layout problem is difficult to solve due to the high-dimensional nature of the problem and the abundance of local minima. International Energy Agency Wind Task 37 has developed reference wind plants to be able to compare wind plant layouts across literature. One study developed a method to reduce the number of design variables to increase the computational efficiency of the wind plant layout optimization problem Stanley and Ning (2019) using a boundary grid method. Other layout optimizations have focused on gradient-based optimization algorithms using analytical gradients and approximations to the model to avoid local minima Thomas and Ning (2018); Stanley et al. (2019). This task becomes even more challenging when addressing multiple technologies at a single site.

Previous work has pointed to the difficulties associated with HPP optimization and sizing and has identified several research opportunities in this area (Dykes et al., 2019). Existing literature focuses on challenges with the system objectives, decision variables, and constraints associated with the system (Upadhyay and Sharma, 2014; Haghi et al., 2017). These studies on HRES optimization have covered traditional optimization methods encompassing dynamic programming, mixed integer linear programming, artificial intelligence methods, hybrid methods, and specifically developed software tools (Musselman et al., 2019; Fischetti and Pisinger, 2018; Gebraad et al., 2017; Ning et al., 2019; Cutler et al., 2017). HPP optimization efforts have predominantly focused on technology sizing or objectives such as reliability, resilience, or downtime optimization, making most problems amenable to mixed integer linear programming. In this paper, we go beyond sizing and present an approach to optimize the physical layout of a wind-solar HPP.

Additionally, many systems studied in the existing literature are independent HRES rather than grid-connected systems. In these independent systems, such as microgrids, energy production profile, resilience, and downtime prevention are indeed more important than the raw cost of energy production. At the commercial and utility scale, however, projects are extremely cost-sensitive, and developers will seek small optimizations (on the order of 1-3%) that provide an increase in plant profitability. This paper focuses on utility-scale wind and solar hybrid plants. Specifically, this work focuses on a simplified layout optimization method for hybrid wind-solar plants, optimizing hybrid plant layouts for AEP. The goal of this work is to create a well-performing solution in a computationally efficient manner without requiring model gradients.

This paper is divided into three subsequent sections and one appendix. Section 2 describes the HPP model used which considers flicker and shading losses caused by turbines shading solar panels in addition to single source plant modeling factors such as turbine wake losses. Section 3 describes the layout optimization methodology and includes two distinct contributions. First, in Section 3.1 we propose parameterization as an effective tool to reduce the complexity and dimensionality of the hybrid layout optimization problem. Second, in Section 3.3 we argue that the evolution strategies (ES) class of derivative-free optimization methods are well-suited to the parameterized hybrid layout problem, and we demonstrate how hard layout constraints can be transformed through parameterization, projection, and finesse into soft constraints amenable to optimization with ES. Next, in Section 4 we contribute experimental results on four test sites, functioning as a proof of concept that our parameterized ES approach is a viable and reliable method for generating optimized layouts with materially increased AEPs and reduce wake, flicker, and GCR losses over baseline layouts. Next, Sections 4.1 through 4.3 interpret the results of our optimization and suggest general design principles for wind-solar HPPs under various conditions. In Appendix A we provide an approach to inspect, interpret, and debug derivative-free optimization approaches in the context of the layout optimization problem. And finally, we have made the source code implementation used to generate the optimized layouts presented here freely available, https://github.com/NREL/HOPP.

The proposed approach shows a viable path for hybrid plant developers to generate spatially efficient, high-performing, and maintainable hybrid plant layouts while using modest computational resources. In this work, we provide a proof of concept that stochastic optimization of low-dimensional parameterized layouts is an effective method for producing efficient hybrid plant layouts. By interpreting optimized layouts under various scenarios, we reveal possible general principles for wind-solar layout optimization.

## 2 Hybrid Plant Model

A component-based modeling approach was used to allow for system design with increased fidelity and flexibility than would be employed for typical planning, policy, or sizing optimization. Analysis of such a model's design trade-offs often use advanced optimization algorithms due to search spaces that might be nonconvex or ill-behaved, for which the derivatives might be difficult to derive or cumbersome to evaluate. The derivative-free approach is well-suited for such problems but requires fast objective functions because these approaches typically require the evaluation of many candidate solutions. The hybrid plant model developed in this work extends the wind and solar models of the System Advisor Model (SAM), a techno-economic tool that combines renewable energy technologies with financial models (Blair et al., 2018). SAM estimates AEP for a given system configuration using reduced-order models, databases of component performance, and loss factors at multiple points along the simulation. The PV models in SAM use solar resource data containing irradiance and meteorological time series to estimate the energy production based on the type of PV modules, how many of and in what configuration these modules are electrically connected into strings, the type of PV inverters, and the geometric layout of the rows or trackers used for adjusting the angle of the panels to follow the sun, among other factors. The wind model in SAM estimates the energy production of a plant by simulating a single turbine based on its power curve, computing wake effects due to the turbine layout, and applying various

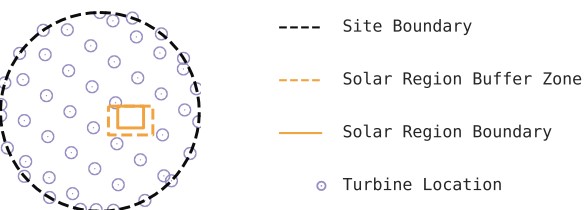

**Figure 1.** An example layout of a wind-solar hybrid power plant inside a circular site boundary.

losses. The financial models can be coupled to the performance models for a variety of ownership structures and markets, allowing financial metrics such as net present value (NPV) to be used as objectives over the energy-based objective in this
paper. The separate wind and solar models are discussed in the following sections.

The shadow flicker model includes the shading interaction between the wind turbines and solar panels using a geometric representation of the turbines to generate time series of shaded portions of the site. At that time step, the shaded areas of the PV module experience some loss of plane-of-array (POA) irradiance, which quantifies the solar power per square meter on the surface of the array and depends on the sun position, the array orientation, ground surface reflectivity and shading. The
effect of shading on the solar panels also include mismatch effects, which are caused by solar cells and modules experiencing different conditions resulting in negative impacts to the output of the entire PV module and string of modules, including power dissipation and heating. The shading-adjusted POA values are used in PVMismatch's two-diode equivalent-circuit model, which represents the current-voltage characteristic of a PV cell under variations in temperature and irradiance, to estimate PV performance and losses relative to unshaded strings (Mikofski et al., 2018; Chaudhari et al., 2018). This shadow flicker model
is simulated for a full year and, to enable fast lookup during the objective evaluation of hybrid AEP, flattened into a table of annual loss factors by location relative to the turbine.

### 2.1 Wind Plant Model

SAM's wind plant model (Freeman et al., 2014) simulates the performance of a wind plant from the wind resource, turbine specifications, and plant layout. The wind resource data, taken from the Wind Integration National Dataset (WIND) Toolkit
(Draxl et al., 2015), is hourly temperature, pressure, wind speed, and wind direction at 80 meters. The turbine used here was SAM's default selection from the turbine library and is 1.5 MW, 77 m in diameter, with max power output at 14 m/s, and it is modeled at a hub height of 80m. SAM's reduced-order wake model options include a Simple (deficit factor-based), the Park (WAsP) and the Eddy Viscosity wake model. The Eddy Viscosity model was selected for its relative robustness and accuracy, and the default turbulence coefficient of 0.1 was used. For each evaluation of the objective function, the wind plant layout is
recalculated for simulating the wake losses and wind AEP.

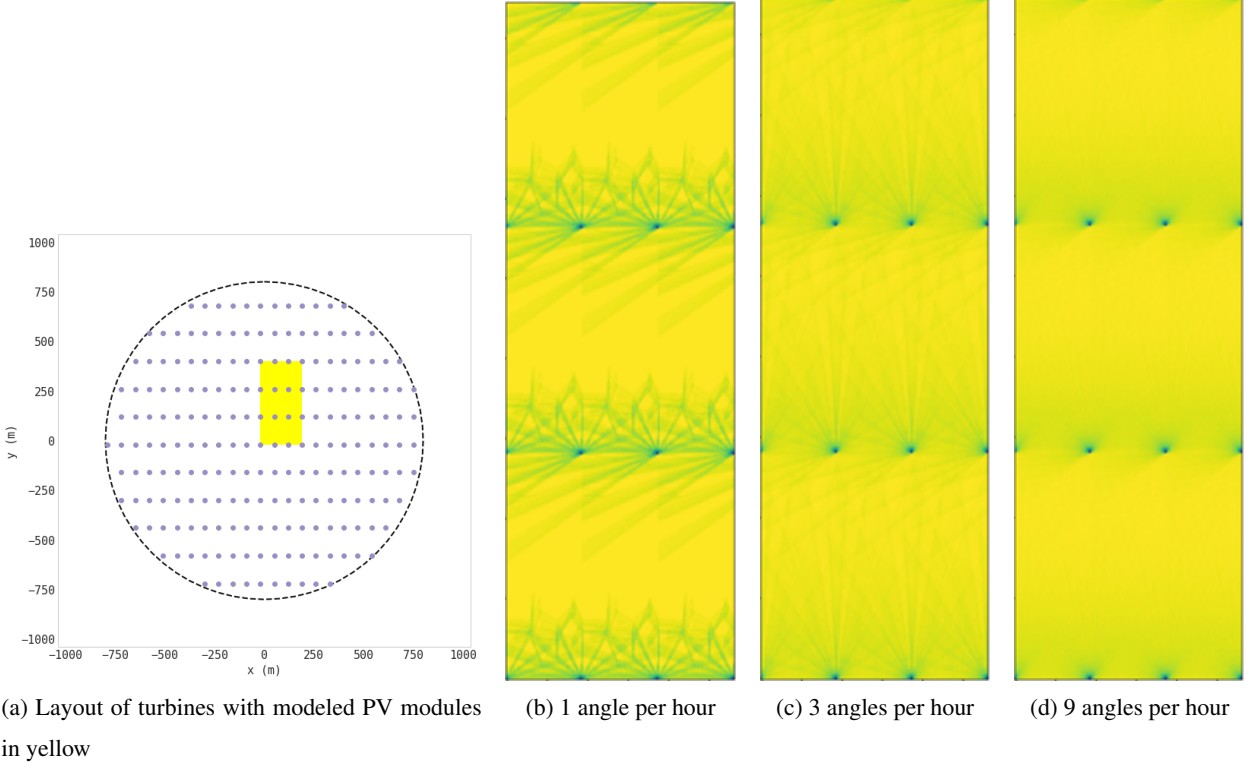

(a) Layout of turbines with modeled PV modules in yellow

(b) 1 angle per hour

(c) 3 angles per hour

(d) 9 angles per hour

**Figure 2.** Heat map of POA loss for a 4x4 turbine grid generated by repeating the central grid cell with different simulation resolutions. (b) and (c) both run three time steps per hour, with 1 or 3 blade angles per time step. The simulation length for each was a few days.

## 2.2 Solar Plant Model

SAM's simple PV plant model, PVWatts®, simulates solar generation using solar resource and high-level system design inputs, such as size, module type, array type, tilt, azimuth, GCR, and DC-to-AC ratio (Dobos, 2014). The solar resource, taken from the National Solar Radiation DataBase (NSRDB), is hourly global horizontal irradiance, diffuse horizontal irradiance, direct normal irradiance, wind speed, and temperature (Sengupta et al., 2018). PVWatts makes simplifying assumptions about the system and array design rather than modeling specific components. For the layout problem here, this is appropriate because the detailed system, array and electrical parameters available in SAM's Detailed PV model would not affect the placement of the solar array within the site, whereas effects due to tracking modes and GCR can still be estimated with the faster simulations of PVWatts. For this hybrid layout optimization, we used a single-axis tracking system, as is found for most utility-scale systems, and fixed all parameters besides GCR and system size.

## 2.3 Shadow Flicker Model

The shadow flicker model uses the turbine dimensions, the site's latitude and longitude, and a PV module model to create a lookup table, or map, of annual loss multipliers by location relative to the turbine. The shadow of the turbine falling on the

xy plane for every time step is calculated from the tower height and width; the turbine's blade length, width, and angle; and
the sun elevation and azimuth. To calculate the shape of the shadow on the ground from a wind turbine with a given radius, we assume a tower height of 2.5R, tower width of R and blade width of R/16, following (Mamia and Appelbaum, 2016). Geometrically, the blade angle is phi, the degrees from the z-axis, whereas the wind direction is theta, where the turbine is pointing towards, with 0 at north, moving clockwise. The three blade shadows are calculated from their positions along the parametric equation of a general ellipse, which represents the shadow of the swept area deformed by the yaw and the sun
position. The number of blade angles to run per time step is an input to the model and is not calculated from wind speed. The position of the output shadow polygon on the ground is relative to the turbine located at (0, 0). The code for these calculations are found in the shadow_flicker.py source file in the HOPP repository, https://github.com/NREL/HOPP/blob/master/hybrid/ layout/shadow_flicker.py. The plane-of-array (POA) irradiance is assumed to be reduced by 0.9 uniformly within the turbine's shadow. To calculate the impact of the reduced irradiance on the PV power output, the model places the turbine among a grid
of 10-module PV strings aligned vertically, where the module is the default PVMismatch configuration of 96-cell, 3-string modules with a bypass diode per string. For each step, the power output of each PV string is calculated using the full POA for unshaded modules and the reduced POA for shaded modules. We did not model temperature effects, such as heat transfer with ambient or resistive heating, due to the partial shading. Further, the PV array could experience shading from multiple turbines, and how these shadows overlay across a single string might result in power loss that has a nonlinear relationship with
the number of shaded modules; however, the simulation time due to additional turbines and an expanded grid of PV strings would be far too long to use within an optimization loop.

Therefore, we investigated ways to reduce the complexity of the model while preserving the required PV power loss information by exploiting the periodicity inherent to each cell of the inner turbine grid, and comparing a map generated from a full simulation with all nearby turbines and one generated by superimposing the losses from a single turbine. Due to the regularity

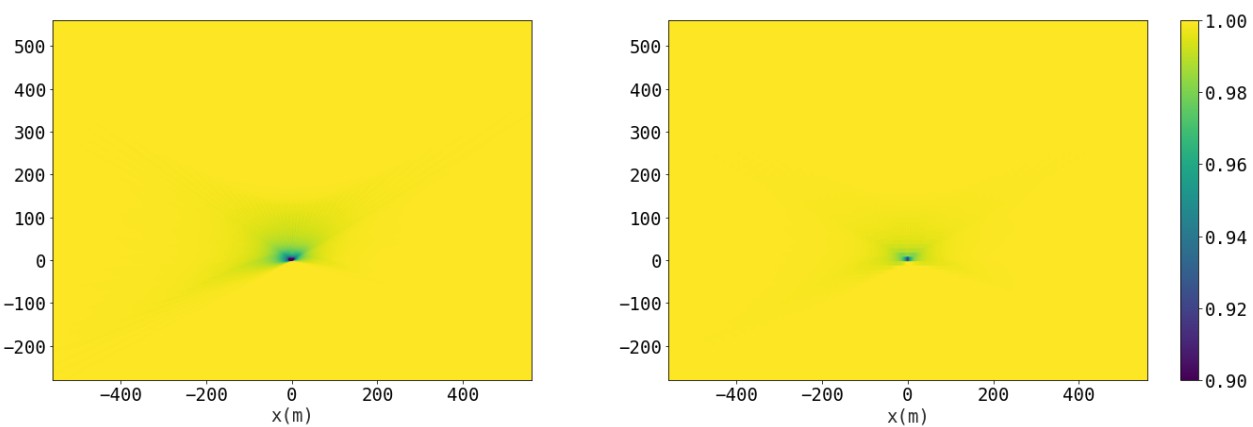

**Figure 3.** Heat maps of POA loss (left) and PV power loss (right) for a year at $(33.209°, -108.283°)$, simulated at 15-minute intervals with 12 blade angles per step for a 77 meter diameter turbine at (0, 0).

of the inner turbine grid the shadow cast upon a cell internal enough to the grid is representative of other internal cells, so rather than simulating the shadows from all the turbines, we used a grid of 4x4 turbines to examine the central grid cell. The output of the shadow model is a 2D map of POA loss due to turbine shadow. The output of the PV power loss model is a map of PV power output loss on a 10 module string basis due to the reduced effective irradiance and module mismatch.

Figure 2 shows the results of piecing together from that central grid cell the image of the 4x4 turbine grid shadows, on a
small demonstration case of only a few days and a few blade angles per hour. In particular, Figure 2b shows clearly how long shadows were cut off due to the 4x4 limitation. As the resolution increases, as shown in Figure 2c and Figure 2d, the effect is less noticeable because the relative weight of the long shadows decrease; similarly, as the simulation length increases, the relative POA loss at sunrise and sunset hours decreases.

The central grid cell's shadows are not representative of cells that are near the edge of the inner grid. To further generalize
the shadow flicker model, we compared the similarity of shadow and PV power loss maps created by the central grid cell and by adding a single turbine's shadow and PV power loss map for each turbine. Figure 3 shows the shadow and PV power loss maps for an area around a turbine at (0, 0) that is 8 turbine diameters to the north, west, and east and 4 turbine diameters to the south at the latitude $33.209°$, longitude $-108.283°$. Weighted throughout the entire year by POA loss and PV power output loss, respectively, the shadow losses range from the greatest shadow reduction of 40.6% to no shadow reduction (100%) far
from the turbine, with an average reduction of 99.95%; whereas the PV power losses range from a reduction of 90.5% to 100%, with an average of 99.97%. Above the POA loss, the additional mismatch losses were found to be minimal. PV power loss at each point is a little less than shadow loss due to averaging across the 10 modules. This result reflects the simple PV module assumptions we made and could change given additional details, such as flicker's effect on temperature and power electronics. Comparison of the central grid cell's shadow and PV power losses to those generated by composing the losses from a single
turbine showed good agreement, with an average normalized difference of 0.176% for shadow and 0.466% for PV power. For a given candidate's PV array dimensions and location, the aggregate flicker loss is the sum of losses from each turbine and is multiplied to the PV power output.

## 3  Optimization Methodology

### 3.1  Parameterizing Hybrid Plant Layouts

Allowing every dimension of a hybrid plant layout to be optimized as a free variable makes for an extremely high-dimensional optimization problem. The position and type of each turbine and solar module must be chosen along with the configuration of the solar module strings. Additionally, layouts generated in this way likely have irregular designs, which can be undesirable for construction, maintenance, cabling, and other purposes. Here, we propose a parameterization that draws inspiration from recent work done to simplify the layout optimization of wind plants (Stanley and Ning, 2019). We propose a parameterization
of hybrid plant layouts that significantly reduces problem dimensionality and constrains the solution space to practical, regular layouts. While projecting the design space into a low-dimensional representation necessarily eliminates many potential layouts, we find that many excellent solutions can be readily discovered within the parameterized search space.

**Table 1.** Optimization variables used in our problem formulation.

| Parameter | Definition | Bounds | | Prior | |
|---|---|---|---|---|---|
| | | Min | Max | $\mu$ | $\sigma$ |
| Boundary spacing | Relative spacing of turbines along boundary (minimum turbine spacing)$(1 + $ boundary spacing$) = $ spacing | 0 | 100 | 5 | 5 |
| Boundary offset | Boundary turbine placement offset as ratio of boundary spacing | 0 | 1 | .5 | 2 |
| Grid angle | Interior turbine grid rotation | 0 | $\pi$ | $\pi/2$ | $\pi$ |
| Grid aspect power | Logarithm of the interior turbine grid aspect ratio $e^{\text{grid aspect power}} = \frac{\text{column spacing}}{\text{row spacing}} = $ grid aspect ratio | -4 | 4 | 0 | 3 |
| Row phase offset | Interior grid turbine row starting offset as a multiple of intra-row spacing | 0 | 1 | .2 | .5 |
| Solar x position | Relative east-west position of the solar region within the site | 0 | 1 | .5 | .5 |
| Solar y position | Relative north-south position of the solar region within the site | 0 | 1 | .5 | .5 |
| Solar aspect power | Logarithm of the solar region's aspect ratio. $e^{\text{solar aspect power}} = \frac{\text{east-west size}}{\text{north-south size}} = $ solar aspect ratio | -4 | 4 | 0 | 3 |
| Solar GCR | Ground coverage ratio of the solar region | .2 | .9 | .5 | .5 |
| Solar southern buffer | Southern solar buffer zone relative to minimum turbine spacing southern buffer length $= $ (solar southern buffer)$(1 + $ minimum turbine spacing$)$ | 1 | 10 | 4 | 4 |
| Solar x buffer | Eastern and western solar buffer zone relative to minimum turbine spacing Southern buffer length $= $ (solar southern buffer)$(1 + $ minimum turbine spacing$)$ | 1 | 10 | 4 | 4 |

Our parameterization, summarized in Table 1, comprises of 11 dimensions: 5 turbine placement parameters and 6 solar placement parameters. The exact implementation mapping parameter values to turbine and solar locations, including placement constraint handling can be found beginning on line 176 of the hybrid_parameterization.py source file in the HOPP repository, https://github.com/NREL/HOPP/blob/master/examples/optimization/layout_opt/hybrid_parametrization.py#L176. Where reasonable, we chose parameters with $(0, 1)$ or similar bounds and relatively smooth, uniform impacts on site layout. Boundary spacing and boundary offset determine the placement of turbines along the site boundary. Boundary spacing determines the distance between turbines placed along the site boundary relative to the minimum turbine spacing, 200 m in our experiments. Boundary offset determines the rotational phase offset along the boundary when placing boundary turbines. Grid angle, grid aspect power, and row phase offset control the angle, aspect ratio (the ratio of intra-row to inter-row spacing), and the placement offset between rows. Perturbing the raw aspect ratio causes relatively small changes to the layout for values larger than 1, moderate changes for values near 1, and large changes for values near 0. Therefore, we optimize over the logarithm of the grid aspect ratio which yields a more linear response. These turbine placement parameters are analogous to those in (Stanley and Ning, 2019), and we direct the reader there for further description of and justification for these parameters.

For solar placement, we consider layouts with a single contiguous solar region that is rectangular in shape except when placed against a site boundary. The center point of the solar region are specified by the solar x-position and solar y-position variables, which range from 0 (along the western and southern bounds of the site, respectively) to 1 (along the eastern and

northern bounds). The aspect ratio of the solar region is determined by the solar aspect power variable, which is equal to the logarithm of the aspect ratio of the east-west and north-south lengths of the solar region. To allow the optimizer to minimize shading and flicker losses, we define two buffer zones around the solar region from which turbines are excluded. The solar southern buffer and solar x buffer specify the size of the southern and east-west buffers beyond the minimum setback, 200 m in our case, as a multiple of the minimum setback. Finally, the GCR of the arrays within the solar region is included as a design variable. Allowing flexible solar placement beyond the southern boundary of a site (where no shading or flicker losses would be incurred) enables the generation of layouts with interior or solar regions which may also have little or no shading or flicker losses, but which allow for greater turbine separation and therefore lower wake losses. In fact, many of the optimized layouts discussed in Section 4.1 make this trade-off.

This parameterization does not specify the size of the solar region nor the spacing of turbines within the inner grid. These two variables are instead determined by performing binary searches to find the least dense layouts that accommodate all nonboundary turbines and all solar modules up to the specified wind and solar capacity constraints. Using a binary search to walk along the constraint boundary increases the layout search efficiency by generating candidate layouts that accommodate the maximum allowed solar and wind capacities given their parameterization. Due to the possible nonconvexity of the site boundary, turbine spacing and solar region size are not generally guaranteed to have monotonic responses to the number of turbines or solar modules, potentially causing a binary search to return suboptimal values; however, we did not encounter any issues in using this approach. Convoluted nonconvex site boundaries might need to be simplified for this approach to work, or a binary search could be replaced with a more robust technique that could reliably handle such conditions.

### 3.2 Objective Design

In this proof of concept, we choose to simply maximize estimated AEP, subject to separate wind and solar nameplate capacity constraints of 75 MW and 50 MW, respectively. As confirmed in Table 4, these capacity constraints were chosen to yield similar solar and wind AEPs of approximately 110 GWh at the high-correlation location. We use up to 50 1.5 MW turbines with a minimum spacing of 200 m between turbines and between turbines and solar modules. Other objectives are possible including capacity factor, net present value, payback time, or carbon payback time. One objective of particular interest for hybrid plants is maximizing utilization of a limited grid interconnect, which demonstrate can be similarly optimized with this approach in section 4.3.

### 3.2.1 Soft Constraints

Derivative-free optimization methods generate candidates from generative distributions that can be difficult to adapt to hard constraints, so instead we use two forms of soft constraints to guide candidate generation to feasible layouts. We penalize only infeasible solutions, leaving the AEP objective fully intact within the feasible region. When evaluating infeasible solutions, we project them onto the nearest feasible solution by clamping parameter values to their bounds. Our first penalty is a simple quadratic penalty for parameter values outside their constraint boundaries. A quadratic penalty allows the optimizer to stray somewhat beyond the boundary, but due to the quadratic nature of the penalty, the optimizer is neatly repelled from highly

infeasible solutions. The second penalty penalizes layouts for which many parameterizations exist due to interference of the site boundary with the solar region's layout. We penalize layouts with excessive solar buffers that extend beyond the site boundary when a smaller solar buffer would result in the same layout. And we penalize layouts with solar aspect ratios that differ from the actual solar region's aspect ratio or that specify a center of the solar region that does not match the actual center of the solar region (as computed from its axis-aligned rectangular bounds). In these cases, we simply add quadratic penalties for these deviations from the ideal parameterization of a given layout, and we did not find it necessary to carefully tune the relative weights of each penalty to get good performance and generate reasonable candidates. The exact implementation details can be found beginning on line 337 of the hybrid_parameterization.py source file in our repository, https://github.com/NREL/HOPP/blob/7ffb8c58d164ea32f2e0267dbe1869ac6fac9201/examples/optimization/layout_opt/hybrid_parametrization.py#L337.

### 3.2.2  Objective Function

Combining the AEP estimate with the soft constraint penalties results in Equation 1, the objective function used in our experiments:

$$\underset{\boldsymbol{x}}{\text{maximize}} \quad f(\boldsymbol{x}) = P_{\text{wind}}(\boldsymbol{x}) + P_{\text{solar}}(\boldsymbol{x})$$

$$- \eta_0 \left\| \max(\boldsymbol{0}, \boldsymbol{x} - \boldsymbol{x}_{\max}) \right\|^2 - \eta_0 \left\| \max(\boldsymbol{0}, \boldsymbol{x}_{\min} - \boldsymbol{x}) \right\|^2 - \eta_1 \sum_{s \in S} \left\| S_s(\boldsymbol{x}) \right\|^2 \tag{1}$$

where:

$\boldsymbol{x}$ is a column vector comprising of the scalar values from Table 1

$P_{\text{wind}}(\boldsymbol{x})$ yields the AEP of the wind plant in the layout described by $\boldsymbol{x}$

$P_{\text{solar}}(\boldsymbol{x})$ yields the AEP of the solar array in the layout described by $\boldsymbol{x}$

$\eta_0$ and $\eta_1$ are soft constraint nuisance parameters, set to $\eta_0 = 0.1$ and $\eta_1 = 1.0$ in our experiments

$\boldsymbol{x}_{\max}$ and $\boldsymbol{x}_{\min}$ are column vectors comprising of the minimum and maximum values from Table 1

$S$ is the set of layout-based soft constraints penalty functions as described in Section 3.2.1

and $S_s(\boldsymbol{x})$ returns the amount by which soft constraint $s$ is violated by $\boldsymbol{x}$.

### 3.3  Optimization Methods

Algorithm 1 lists an outline of the evolution strategies (ES) approach to stochastic optimization. Evolution strategies is a good fit for the hybrid plant layout optimization problem due to the highly nonconvex objective function, the difficulty in obtaining derivatives, their potentially noninformative nature, and the ability to generate multiple good layouts for consideration. Some evolution strategy implementations simply return the mean or other measures of $\mathcal{G}$. We chose to instead return the best solution found, $\mathbf{c}^*$, which experimentally improved performance over returning the mean in every comparison we tested. We evaluated three ES-based approaches for optimizing hybrid plant layouts.

---

**Algorithm 1:** An evolution strategies framework for stochastic optimization.

---

$\mathcal{G} \leftarrow \mathcal{G}_0$                                                     `// Initialize generative distribution`

$\mathbf{c}^* \leftarrow \varnothing$                                                     `// Set current best solution to none`

**while** *optimizing* **do**

    Draw candidate population, $P = \{\mathbf{c}_1, ..., \mathbf{c}_\lambda\}$, from $\mathcal{G}$

    Obtain the objective value of each candidate $F(\mathbf{c}_i)$

    $\mathbf{c}^* \leftarrow \underset{\{\mathbf{c}^*\} \cup \{\mathbf{c}_1, ..., \mathbf{c}_\lambda\}}{\mathrm{argmax}} F(\mathbf{c}_i)$                    `// Update best candidate found`

    Update $\mathcal{G}$ using candidate evaluations $\{(\mathbf{c}_1, F(\mathbf{c}_1)), ..., (\mathbf{c}_\lambda, F(\mathbf{c}_\lambda))\}$

**return** *$c^*$*

---

### 3.3.1 Random Search

Random search (RS) is a straightforward evolution strategy where $\mathcal{G}$ is fixed and never updated. RS simply generates candidate layouts from a stationary distribution, keeping track of the best-performing layout found so far; thus, RS provides a simple interpretable baseline. When the search is terminated, the best layout is returned. In our experiments with RS, we chose $\mathcal{G} \sim \mathcal{N}(\mu, \mathrm{diag}(\sigma))$. Due to its static nature, RS's efficiency depends on the performance of candidate solutions drawn from $\mathcal{G}$. Unfortunately, this is a difficult task because good prior distributions are those that have a high probability of generating good layouts, but if we knew what parameter values would yield good layouts, we would not need to search for them.

### 3.3.2 Cross-Entropy Method

The cross-entropy method (CEM) is an evolution strategy that originates from rare event simulation that has been adapted to both discrete and continuous variable optimization problems (Y. Rubinstein, 1997; de Boer et al., 2005). We used the common multivariate Gaussian form of CEM optimization, outlined in Algorithm 3.3 of (de Boer et al., 2005), with $\mathcal{G}_0 \sim \mathcal{N}(\mu, \mathrm{diag}(\sigma))$, a population size of $\lambda = 200$, and a selection proportion of $\gamma = \frac{1}{3}$. These choices mean that 200 candidates were generated on each iteration of Algorithm 1, and $\mathcal{G}$ was set to the maximum likelihood multivariate Gaussian fit of the best-performing (highest valued) 67 of those candidate layouts. CEM is an effective strategy that does a good job of efficiently finding high-performing layouts, but it can be prone to getting stuck in local maxima.

### 3.3.3 Covariance Matrix Adaptation Evolution Strategy

Covariance matrix adaptation evolution strategy (CMA-ES) (Hansen and Ostermeier, 1996, 1997) is a sophisticated evolution strategy that augments CEM's approach with several techniques to better avoid local minima and to update the covariance matrix of $\mathcal{G}$ in a way that is analogous to the approximation of the inverse Hessian matrix as in quasi-Newton methods, such as the Broyden–Fletcher–Goldfarb–Shanno algorithm (BFGS) (Fletcher, 2000), the limited memory BFGS algorithm (BFGS-L) (Malouf, 2002; Andrew and Gao, 2007), and the earlier Broyden's method (Broyden, 1965). CMA-ES has been extended and

enhanced over the years to increase the algorithm's recombination efficiency (Hansen and Ostermeier, 2001), improve the time complexity of the update step (Hansen et al., 2003), increase robustness in the face of multimodal objective functions (Hansen and Kern, 2004), and more. We applied the modern implementation of CMA-ES as described in Appendix C of *A Tutorial on the Cross-Entropy Method* (Hansen, 2016). As with CEM, we use a prior distribution of $\mathcal{G}_0 \sim \mathcal{N}(\mu, \mathrm{diag}(\sigma))$, a population size of $\lambda = 200$, and a selection proportion of $\frac{1}{3}$.

## 4 Experimental Results

**Table 2.** Comparison of the two test locations.

| Site | Region | Latitude | Longitude | Elevation | Pearson correlation coefficient |
|---|---|---|---|---|---|
| High-Correlation | Central Valley of California | $36.334°$ | $-119.769°$ | $70m$ | $0.28$ |
| Low-Correlation | Southwest New Mexico | $33.209°$ | $-108.283°$ | $2000m$ | $-0.30$ |

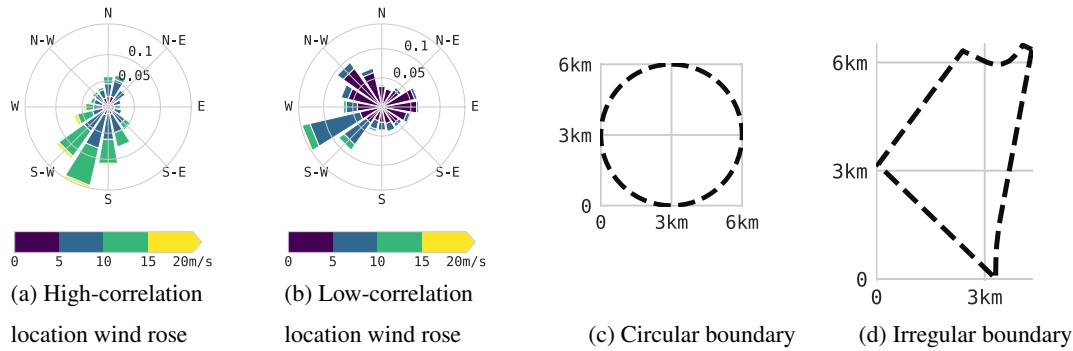

(a) High-correlation location wind rose

(b) Low-correlation location wind rose

(c) Circular boundary

(d) Irregular boundary

**Figure 4.** Wind roses and boundaries for the two locations and two boundaries used in our experiments.

As a proof of concept, we present experimental results generated by applying the proposed hybrid layout optimization approach to the four distinct combinations of two site locations and two site boundaries. We choose two distinct locations outlined in Table 2 in the continental United States having the highest and lowest Pearson correlation coefficient (Pearson and Henrici, 1896) between wind and solar resource, using the resource databases mentioned above. We chose to use the Pearson correlation coefficient because it is the most popular criteria for analyzing the relationship between wind and solar resource (Jurasz et al., 2020; Iwanowski, 2018; Zhang et al., 2013), and unlike Spearman's rank correlation coefficient (Spearman, 1904), it is well-suited to the continuous-valued time-series data used to compare locations. The high-correlation location, in which wind and solar resources tend to be present together with a correlation coefficient of 0.28, is located in California's Central Valley, directly south of Fresno and north of the City of Lemoore, at latitude $36.334°$, longitude $-119.769°$, and an elevation of $70m$. Given this moderate positive correlation coefficient, wind and solar resources in even the highest correlation location in the continental United States complement each other somewhat and, therefore likely to yield increased grid resilience and stability through increased consistency in energy production. The high-correlation location has a predominant wind direction,

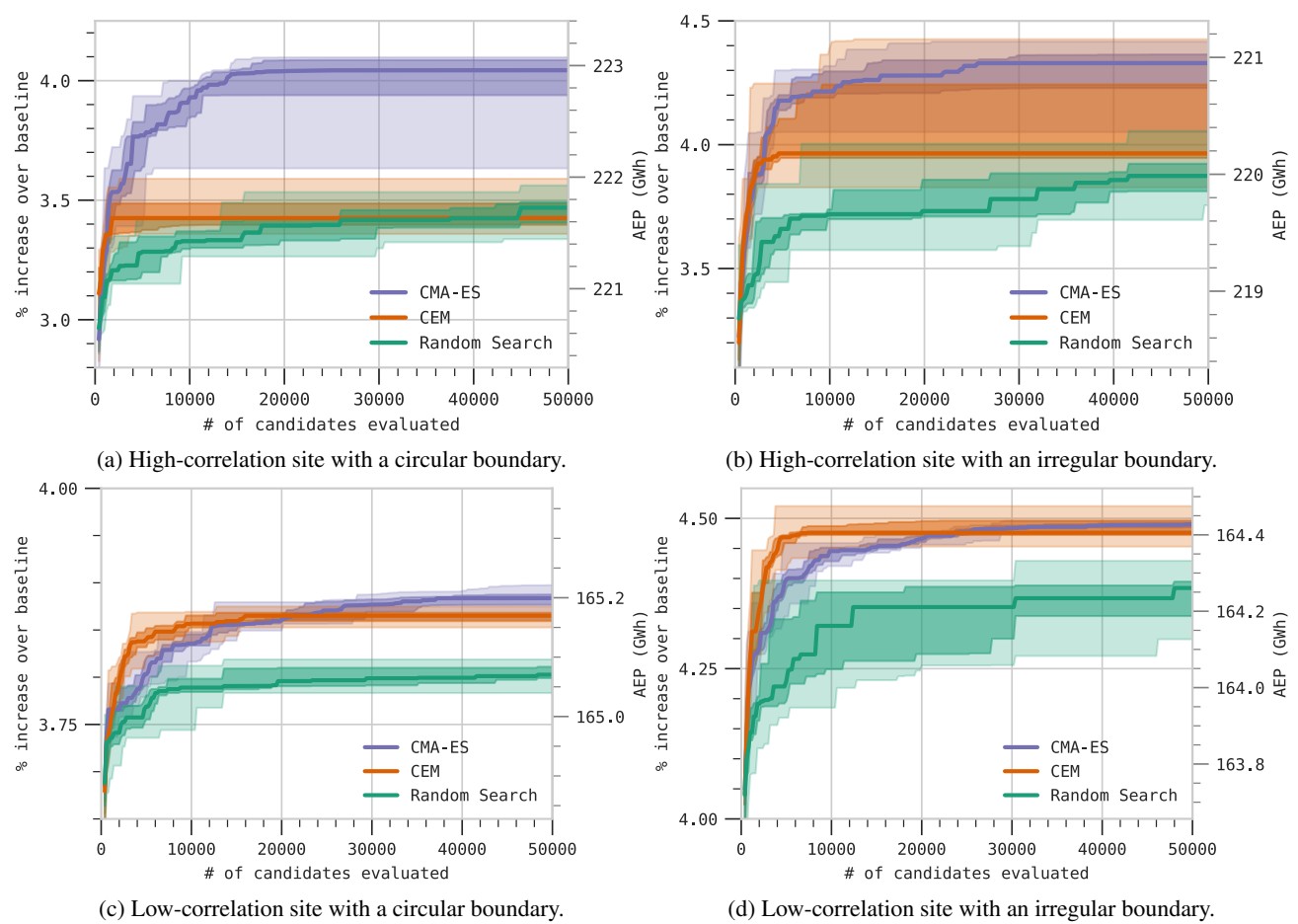

(a) High-correlation site with a circular boundary.

(b) High-correlation site with an irregular boundary.

(c) Low-correlation site with a circular boundary.

(d) Low-correlation site with an irregular boundary.

**Figure 5.** Optimization progress curves for each of the three evolution strategies optimization algorithms on each combination of the two site locations and the two site boundaries over the course of 10 optimization runs. Dark lines indicate median values as observed over 10 optimization runs. The dark fill around the median spans the 25th–75th percentile range, and the lighter fill spans the minimum to maximum range.

**Table 3.** Mean performance gains over the baseline site for 10 runs of each optimization algorithm.

| Algorithm | High-Correlation Location | | Low-Correlation Location | |
|---|---|---|---|---|
| | Circular Boundary | Irregular Boundary | Circular Boundary | Irregular Boundary |
| CMA-ES | **3.97%** | **4.30%** | **4.49%** | **3.88%** |
| CEM | 3.45% | 4.08% | 4.48% | 3.86% |
| RS | 3.45% | 3.89% | 4.37% | 3.80% |

as shown in Figure 4a. The low-correlation location is located in Southwest New Mexico, with a latitude of $33.209°$, longitude of $-108.283°$, and an elevation of $2000m$, and it has a resource correlation coefficient of $-0.30$. This location presents wind and solar resources that are typically complementary and therefore present an excellent opportunity for hybrid power generation. As shown in Figure 4b, the low-correlation location's wind direction distribution is more dispersed, with lower typical wind speeds than found at the high-correlation location. In this section, we analyze experimental runs of our layout optimization approach as applied to each of the four combinations of these two locations and each of the two site boundaries: a simple circular boundary with a 3 km radius, as shown in Figure 4c, and a nonconvex wedge-shaped irregular boundary, as shown in Figure 4d.

Table 3 summarizes the results of running each of the three evolution strategies optimization algorithms on each combination of the two site locations and two site boundaries. In all cases, CMA-ES achieved a higher mean performance than either CEM or RS, whereas CEM took second place or tied with RS on each site. Taking a closer look, the optimization progress curves shown in Figure 5 indicate that CMA-ES typically achieves good performance with less variability than the other two methods, but CEM sometimes takes an early lead over CMA-ES which CMA-ES overcomes only after 20k–25k candidates have been evaluated. Random search has poor performance overall and tends to have a higher inter-run variability in the performance of its layouts; however, in the high-correlation location with the circular boundary, we see that CEM rapidly becomes stuck in a local maxima, and RS can eventually outperform CEM in this case. In fact, we see that CEM rapidly finds a local maxima in all four cases, and this is likely why CMA-ES, which is more robust to local maxima, is able to eventually beat CEM in these tasks. Interestingly, we find that on every test site, most gains are achieved by the first or second iteration of each of the three algorithms. Among the 200 randomly generated initial candidates, there was always a site that increased the objective value by $2.8$ to $4.0\%$. That is to say that simply drawing one random generation of candidates from the prior distribution and choosing the best-performing layout from that set yielded the most gains to be had when optimizing layouts using this parameterization. It is possible that, in a more general sense, many layouts could be improved significantly by simply generating a few hundred random perturbations of the layout parameters and choosing the best candidate found; however, in every case, all three optimization algorithms were also able to squeeze out additional performance beyond this initial improvement, with CMA-ES yielding the best overall results. We also see that CMA-ES and RS continue to eek out additional gains between 40k and 50k evaluated candidates, suggesting that longer runs would likely yield additional gains. These results are not meant to be a definitive examination of which approach is best for the hybrid layout problem, but they are instead meant to show that there are viable evolution strategies-based approaches to solving the hybrid layout problem. It is possible that with careful tuning, for example, adjusting CEM's convergence parameters, these results would change somewhat; however, we found that CMA-ES was significantly easier to work with and easier to get running than other techniques, and therefore we examine it in more detail later.

## 4.1 A Closer Look at the Generated Layouts

Figures 6–9 show a sampling of solution layouts generated by CMA-ES using our hybrid layout parameterization. Each layout's performance statistics are listed in Table 4. In Figure 6, the high-correlation location and circular boundary generates a diversity

of high-performing layouts. All these layouts pack all or all but one turbine into two inner grid rows, typically aligning turbine rows to an angle at a few degrees offset from the prevailing wind direction. This arrangement minimizes mean wake losses in our eddy viscosity-based wake loss simulation, causing wakes to fall just to the side of downstream turbines under most wind conditions. We also see some solutions, such as the layout shown in Figure 6c, that align the grid closer to perpendicular to the prevailing wind direction. This configuration is also competitive, but the closer spacing between rows in the wind direction results in the southerly turbines incurring a bit more wake losses. Similarly, the solver finds a variety of good solar placements, many of which are nonintuitive, including placements such as those shown in Figure 6c and Figure 6e, which place the solar region along the northern boundary of the site. Despite this northerly placement, the optimizer identified turbine placements that eliminate flicker losses.

Figure 7 shows solutions for the irregular boundary on the same high-correlation location. Unsurprisingly, these solutions share design characteristics with those using a circular boundary, but results differ in a few ways. The "taller" north-south aspect of the irregular boundary causes the optimizer to find solutions that align two and occasionally three rows of turbines with the

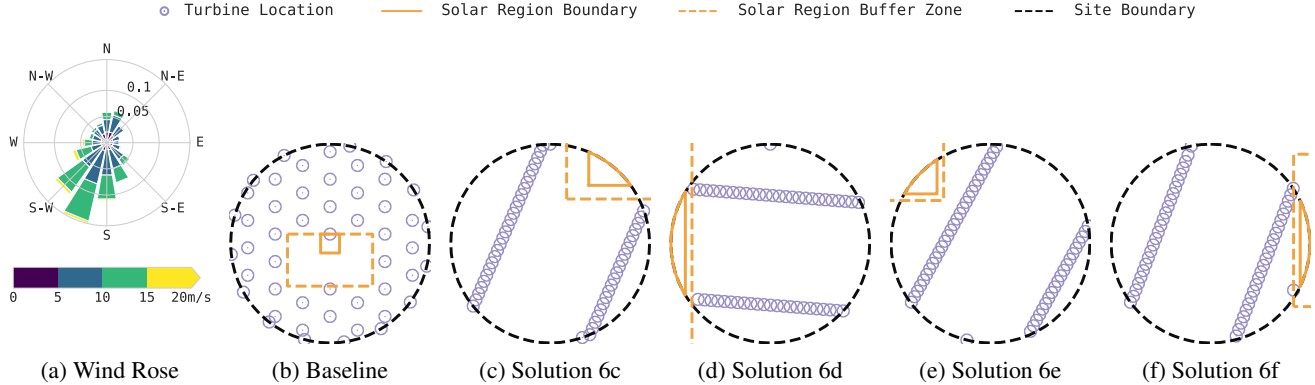

**Figure 6.** CMA-ES solutions for the high-correlation location and circular boundary. Turbine locations are marked in purple, the solar region is drawn with an orange solid line, and the surrounding solar buffer zone is marked with a dashed orange line.

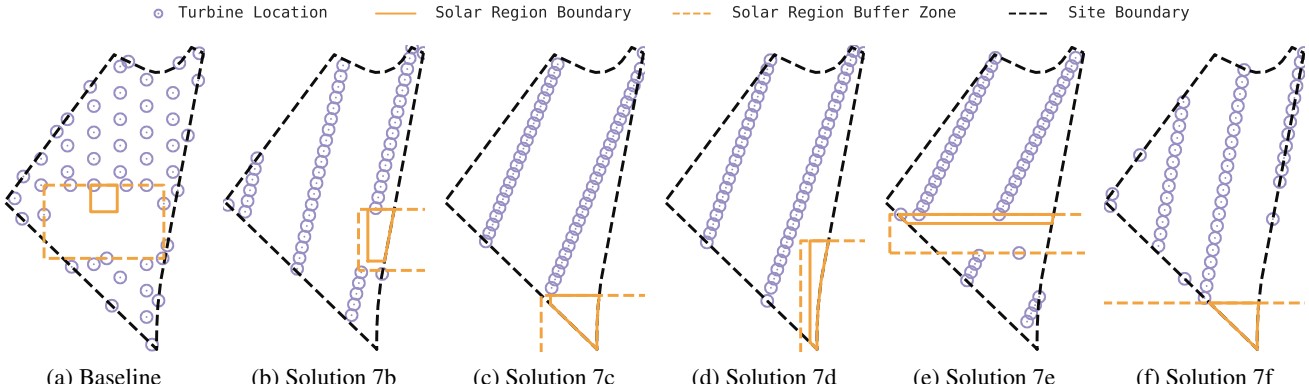

**Figure 7.** CMA-ES solutions for the high-correlation location and irregular boundary.

**Table 4.** CMA-ES layout performance statistics for each solution in Figures 6–9.

| Location | Site | Solution | AEP (GWh) | | | Losses | | |
|---|---|---|---|---|---|---|---|---|
| | | | Total | Solar | Wind | Wake | GCR | Flicker |
| High-Correlation | Circular | Baseline | 214.29 | 101.36 | 112.93 | 4.15% | 6.55% | 0.09% |
| High-Correlation | Circular | 5c | 223.06 | 107.46 | 115.60 | 1.89% | 1.02% | 0.00% |
| High-Correlation | Circular | 5d | 222.08 | 107.46 | 114.62 | 2.72% | 1.02% | 0.01% |
| High-Correlation | Circular | 5e | 222.73 | 107.46 | 115.26 | 2.18% | 1.02% | 0.00% |
| High-Correlation | Circular | 5f | 222.76 | 107.46 | 115.30 | 2.15% | 1.02% | 0.00% |
| High-Correlation | Irregular | Baseline | 211.78 | 101.36 | 110.43 | 6.28% | 6.55% | 0.10% |
| High-Correlation | Irregular | 6b | 220.94 | 107.45 | 113.48 | 3.69% | 1.02% | 0.01% |
| High-Correlation | Irregular | 6c | 220.98 | 107.46 | 113.52 | 3.66% | 1.02% | 0.00% |
| High-Correlation | Irregular | 6d | 221.04 | 107.46 | 113.58 | 3.61% | 1.02% | 0.00% |
| High-Correlation | Irregular | 6e | 220.65 | 107.36 | 113.29 | 3.85% | 1.02% | 0.10% |
| High-Correlation | Irregular | 6f | 220.36 | 107.46 | 112.90 | 4.18% | 1.02% | 0.00% |
| Low-Correlation | Circular | Baseline | 159.02 | 103.57 | 55.45 | 5.38% | 6.47% | 0.08% |
| Low-Correlation | Circular | 7c | 165.21 | 109.09 | 56.12 | 4.24% | 1.57% | 0.00% |
| Low-Correlation | Circular | 7d | 165.16 | 109.09 | 56.07 | 4.32% | 1.57% | 0.00% |
| Low-Correlation | Circular | 7e | 165.22 | 109.09 | 56.13 | 4.22% | 1.57% | 0.00% |
| Low-Correlation | Circular | 7f | 165.20 | 109.09 | 56.10 | 4.26% | 1.57% | 0.00% |
| Low-Correlation | Irregular | Baseline | 157.36 | 103.58 | 53.79 | 8.21% | 6.47% | 0.08% |
| Low-Correlation | Irregular | 8b | 164.40 | 109.07 | 55.34 | 5.57% | 1.57% | 0.02% |
| Low-Correlation | Irregular | 8c | 164.43 | 109.06 | 55.37 | 5.50% | 1.57% | 0.03% |
| Low-Correlation | Irregular | 8d | 164.43 | 109.06 | 55.37 | 5.50% | 1.57% | 0.03% |
| Low-Correlation | Irregular | 8e | 164.44 | 109.06 | 55.38 | 5.50% | 1.57% | 0.03% |
| Low-Correlation | Irregular | 8f | 164.44 | 109.06 | 55.38 | 5.50% | 1.57% | 0.03% |

longer chords of the boundary, again offsetting turbine rows a bit from the prevailing wind direction. Unlike with the circular boundary, some solutions place a smattering of turbines along the site boundary, taking advantage of the additional breathing room afforded by this boundary. In most cases, the solar is packed into the southern tip of the site, eliminating flicker losses entirely; however, a few competitive layouts were found that place the solar region deep in the site's interior, an interesting trade-off that increases turbine spacing at the cost of some flicker and shading losses.

The solutions shown in Figure 8 are generated layouts for the low-correlation location and circular boundary. Here, we see that the more uniform and lower speed wind distribution results in very different solutions than at the high-correlation location. In response to a less concentrated wind direction distribution, the solver proposes layouts that space turbines evenly and place the solar region near the site center, giving turbines some additional separation. Similar results are shown in Figure 9 using the irregular boundary, which primarily differ in an increased utilization of boundary turbines, and placement of the solar region into the northeastern corner of the site. These solutions are likely found because placing the solar in this corner actually causes boundary turbines to avoid the corner and therefore achieve increased spacing. A further-refined parameterization might specially handle border turbine placement in sharp boundary peninsulas such as this one.

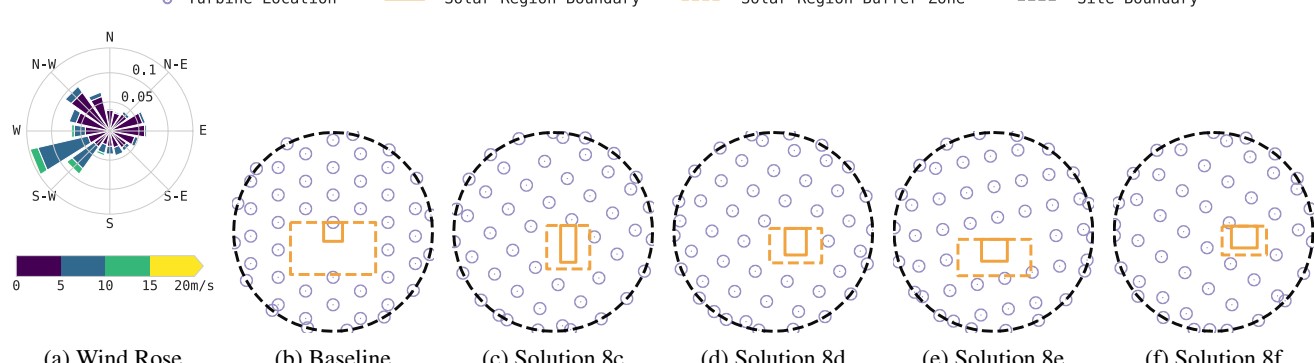

(a) Wind Rose    (b) Baseline    (c) Solution 8c    (d) Solution 8d    (e) Solution 8e    (f) Solution 8f

**Figure 8.** CMA-ES solutions for the low-correlation location and circular boundary.

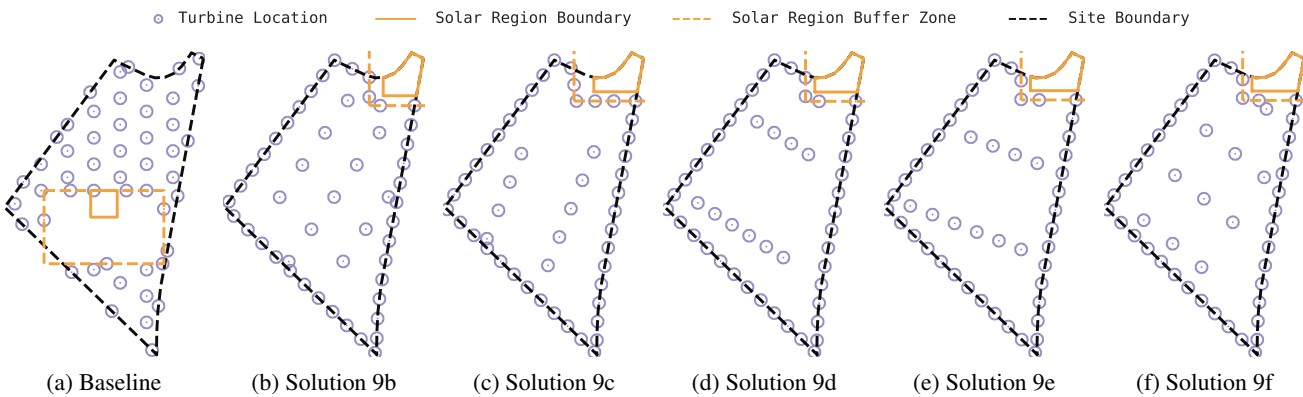

(a) Baseline    (b) Solution 9b    (c) Solution 9c    (d) Solution 9d    (e) Solution 9e    (f) Solution 9f

**Figure 9.** CMA-ES solutions for the low-correlation location and irregular boundary.

Table 4 reveals that solutions to the high-correlation scenarios have approximately an order of magnitude greater spread in AEP than the low-correlation solutions, and this difference stems almost entirely from differences in wind-generated production. Curiously, the high-correlation sites produce only approximately twice as much wind energy as the low-correlation sites, not nearly enough to explain the much larger difference in AEP. It is possible that higher resource correlation presents a more challenging optimization objective, partly due to the greater impact of flicker losses on solar AEP. It is more important to avoid panel flicker when solar generation is high and under high-correlation conditions solar generation is high when wind generation is also high, causing shading turbines to inflict greater flicker losses on solar AEP. This proposition is supported somewhat by the overall greater flicker losses seen in high-correlation solutions, but more investigation is needed to fully understand the cause of the variability.

The ability to generate multiple competitive alternative layouts is a distinct advantage of evolution strategies and other stochastic optimization approaches. Here, we see the creative power of these solution methods in finding a large diversity of viable candidate layouts, all of which yield high objective function scores. In choosing to lay out a hybrid site, one might use these methods to generate a number of good candidate sites, and then choose among them based on other important factors

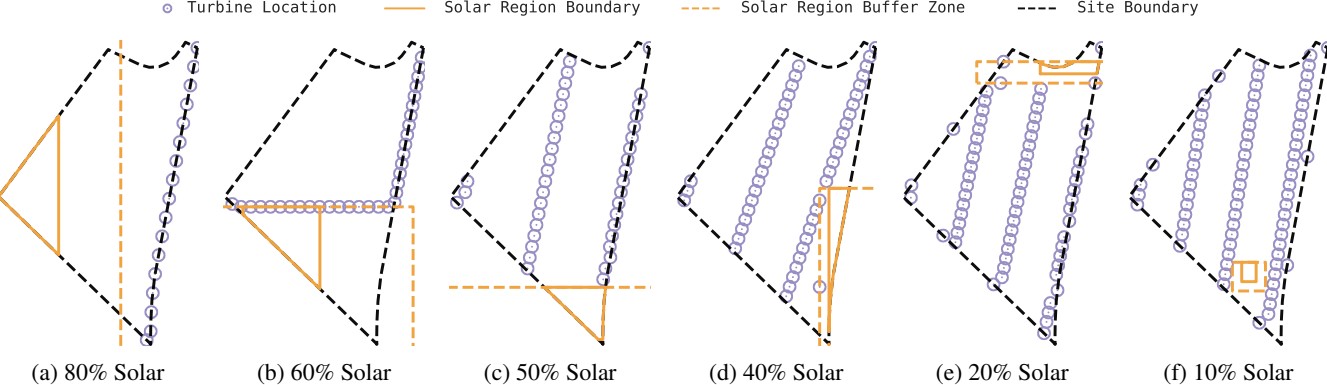

**Figure 10.** Solutions generated for the high-correlation location and irregular boundary for a range of solar and wind capacity mixes, holding total nameplate capacity at 125MW.

that are difficult to encode in such an objective function, such as ease of access, maintenance or cabling concerns, aesthetics, and more.

### 4.2 Layouts for varying capacity mixes.

Figure 10 shows solutions for various solar to wind generation capacity proportions while holding total capacity equal to 125 MW. For solar-heavy specifications, turbines are placed where they will never shade the solar region, and are also spread out to minimize GCR losses, with reducing wake losses only a secondary concern. Figure 10b is a surprising layout which uses the solar region to position the turbines along two rows in a way which also yields low 2.27% wake losses for this location. As solar capacity is decreased and wind capacity is increased, the solar region naturally shrinks and is gradually placed to

allow for reduced wake losses, with solar losses taking a back seat. Figure 10f shows a primarily wind-based layout with solar stuffed in-between two turbine rows almost as an afterthought. However, even in this case flicker losses are only 0.1% and the panels are rarely shaded. These solutions suggest that solar-focused HPPs such as Figures 10a and 10b can maintain high solar production by placing panels near the southern boundary of the site, or in a location away from turbine shading, while leaving ample space for low wake loss turbine placement. Sites with balanced production, such as in Figures 10b, 10c, and

10d are likely well-served by placing solar production along one of the site boundaries, particularly the southern boundary, leaving panels largely unshaded while still providing large contiguous spaces for turbine placement. And figures 10e and 10f suggest that sites using principally wind generation can often find sufficient space between turbine rows to place modest solar generation zones without incurring significant shading or flicker losses, and without moving turbines from their ideal placement. Future work could utilize this layout optimization strategy to, considering the physical layout, identify the mix of

solar and wind generation that optimizes figures of interest such as levelized cost of energy or net present value for a particular site.

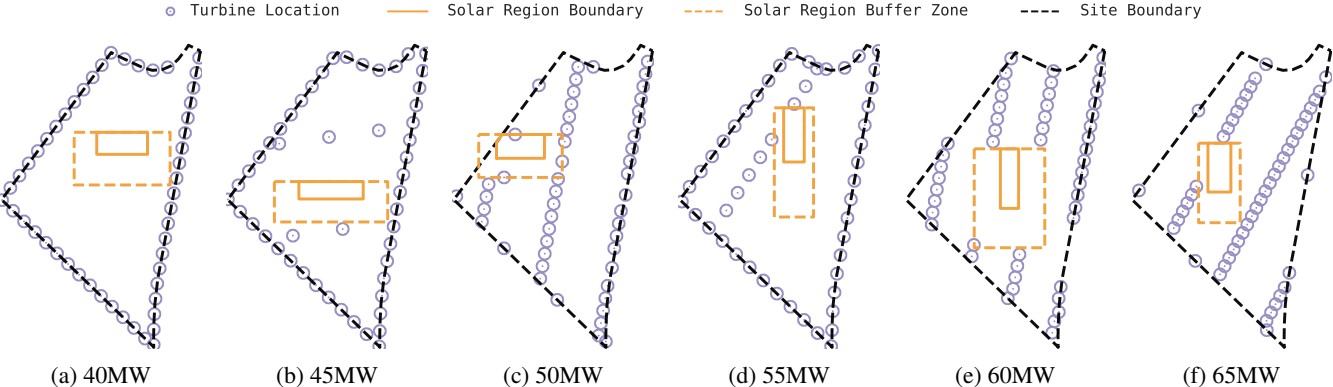

**Figure 11.** Solutions generated for the high-correlation location and irregular boundary for a range of interconnect capacities, maximizing mean interconnect utilization instead of AEP.

## 4.3 Optimizing Alternate Objectives: Interconnect Utilization

To evaluate the flexibility of the parameterized layout optimization approach, we generated the layouts shown in Figure 11, maximizing interconnect utilization instead of AEP for a range of interconnect capacities for high-correlation location and
irregular boundary. Because low interconnect capacities do not realize the benefit of peak energy production, the effect of losses during peak production times is unimportant. Therefore, as the interconnect capacity increases, turbines are shifted from the boundary to the interior grid, reflecting the increased importance of minimizing wake losses when energy production is high. As the interconnect capacity rises above peak production levels, the optimized layouts for 60 and 65MW become similar to those found by maximizing AEP in Figure 7. These results suggest that in highly interconnect constrained scenarios, such as
Figure 11a, hybrid sites may be best served by placing all turbines along the boundary, and solar in a central, unshaded location. This layout type is conducive to maintaining some level of energy production, even in atypical conditions, including rare wind speeds or directions that would cause pathological loss cases under layouts optimized for common conditions. While wake losses will typically be greater under average conditions than with denser rows of turbines, boundary placement, at least with the irregular site, is more robust to atypical conditions. For sites with few straight-line boundaries, such as a rectangular site,
it is likely that setting some turbines back from the boundary by varying amounts would produce similar levels of robustness. For sites with moderate interconnect constraints, such as in Figure 11c and 11d, interior solar placement is still a successful strategy as long as turbines are excluded from shading positions, and a moderate portion of turbines are placed in interior rows. Such layouts maintain reasonable robustness to atypical conditions while taking some advantage of common conditions. Future directions include further incorporation of interconnect design parameters into more complex objective functions would
strengthen and deepen these design guidelines.

# 5 Conclusions

HPP optimization research has focused on system sizing. In this work, we deepen HPP optimization by presenting a practical approach to optimizing not just component sizes, but the physical layout of a wind-solar HPP. And, this framework can be refined and extended to optimize additional design parameters and achieve more detailed objectives as desired.

The proposed HPP layout optimization approach consists of four distinct contributions. First, we presented a modeling for estimating shading and flicker losses incurred due to turbine shading of solar panels, a critical piece for enabling wind-solar layout optimization. Second, we proposed utilizing parametric approach to layout optimization for HPPs in order to reduce the dimensionality of the layout problem and to make it more amenable to non-convex optimization techniques. Third, using a specific parameterization for wind-solar layout optimization we demonstrated the viability of this approach by using ES-based optimizers to generate high-performance layouts. Finally, we analyzed the optimized layouts under a number of scenarios to propose potential general layout guidelines for wind-solar layout optimization.

Future work includes expanding the parameterization to include additional design parameters such as wind and solar capacity mix, turbine type, and site size and shape; adding more detailed objective functions such as NPV and internal rate of return; and accounting for land use restrictions and costs. Additionally, one could formulate more efficient and capable optimization algorithms, including non-evolutionary approaches. The objective function could be improved to account for factors such as cabling, interconnect, maintenance costs, land use restrictions, and budgets. Other improvements are also possible including eliminating capacity constraints and allowing the algorithm to trade between wind turbines and solar modules. Similarly, the objective could be modified to generate layouts that improve existing sites by determining the best locations for additional turbines and solar modules. The specific parameterization presented serves as a starting point that can be extended and adapted to meet the needs of different decision makers, site types, and objectives. This approach opens a viable path for hybrid plant developers to easily generate efficient, maintainable, and aesthetically pleasing layouts using modest computational resources.

## Appendix A: Peering Into The Black Box: Interpreting and Debugging Derivative-Free Approaches

In this appendix, we make a case study of the application of CMA-ES to the high-correlation location and irregular boundary layout problem. We show how examining variable trajectories over the optimization run can give insight into the operation of the optimizer, and can help users understand and debug its performance. Graphing solution losses over an optimization run shown in Figure A1b indicates that the RS of the first iteration immediately finds a configuration that minimizes GCR losses; this corresponds to the GCR trajectory shown in Figure A2, where a low GCR is immediately found to minimize GCR losses. Over the next 200 to 800 evaluations, the optimizer concentrates on finding configurations that reduce or eliminate flicker losses. In most cases, the optimizer found solutions that completely eliminated flicker losses. Finally, the optimizer gradually whittles away wake losses. No configuration here can eliminate wake losses, but the optimizer adjusts the turbine grid position, angle, and aspect ratio, significantly reducing wake losses. During this time, we observed a small amount of variability in flicker losses as the optimizer found that it might be able to trade a bit of flicker loss to reduce wake losses, and we see that a handful of solution layouts, such as Figure 7e, make this trade-off.

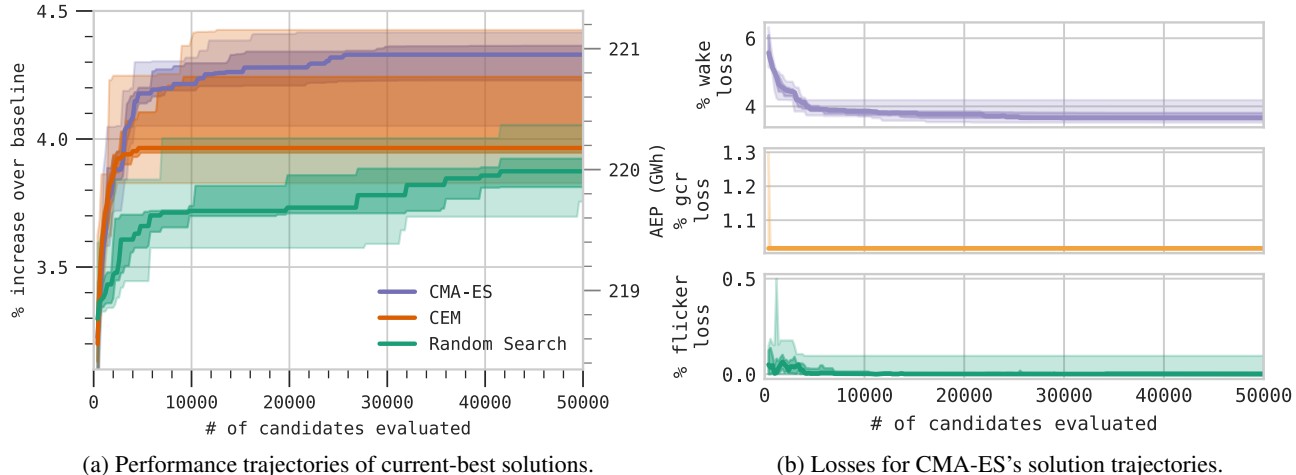

(a) Performance trajectories of current-best solutions.      (b) Losses for CMA-ES's solution trajectories.

**Figure A1.** Optimization progress curves on the high-correlation site with an irregular boundary over 10 optimization runs. Dark lines indicate median values as observed over 10 optimization runs. The dark fill around the median spans the 25th–75th percentile range, and the lighter fill spans the minimum to maximum range.

Analyzing the optimization variables of CMA-ES's solution trajectories shown in Figure A2 indicate that most solutions
use a small boundary offset, moderate boundary spacing, a northeasterly turbine grid angle, and tightly packed turbine rows. Solution solar configurations generally had moderate (near-square) aspects, although some such as the layout shown in Figure 7e are wide along the east-west (x) axis and narrow along the north-south (y) axis. Given ample space to place the solar capacity, low GCR's were universally preferred by CMA-ES, minimizing internal shading (GCR) losses. Similarly, the optimizer universally finds that large east and west buffer regions around the solar are not required to reduce flicker. From the x
(east-west) and y (north-south) solar position trajectories, we see that many solutions sensibly pack the solar region into the southeast corner of the site. Placing the solar at the southern end of the site eliminates turbine shading and flicker on the solar, but it can also pack turbines closer together into the northern portion of the site; however, other good placements are found by the optimizer, including as shown in Figure 7e, which places the solar closer to the middle of the site and uses a southern buffer to reduce shading and flicker losses. This alternative arrangement is competitive because it allows the turbines to be spaced
farther apart, helping to reduce wake losses. This trade-off would be more salient when using larger solar generation capacities and/or smaller wind capacities, causing the solar region to consume more space relative to wind turbines.

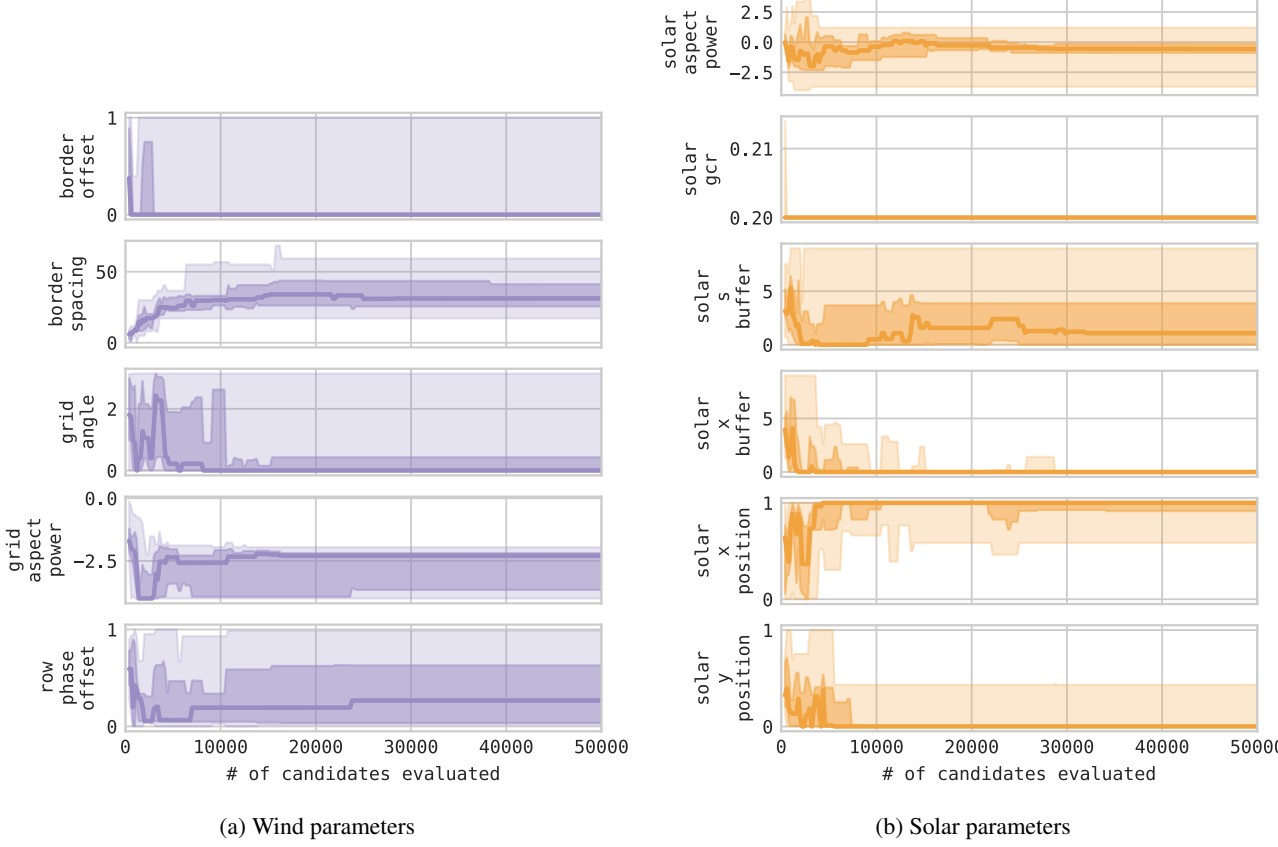

|     |     |
| :-: | :-: |
| (a) Wind parameters | (b) Solar parameters |

**Figure A2.** Solution trajectories using CMA-ES on the high-correlation site and irregular boundary. Dark lines indicate median values. The dark fill around the median spans the 25th–75th percentile range, and the lighter fill spans the minimum to maximum range observed over 10 optimization runs.

*Code availability.* The experimental site optimization code used in this paper is available under the BSD-3-Clause open source licence on GitHub as the NREL Hybrid Optimization and Performance Platform (HOPP) project, https://github.com/NREL/HOPP, Tripp et al. (2020).

*Author contributions.* Jennifer King supervised and managed the project in which this research was accomplished. Darice Guittet developed
and implemented the objective function, shading model, and flicker models. In doing so, she integrated the optimizer with the System Advisor Model Blair et al. (2018). Aaron Barker's work to map hybrid power production potential was used to identify sites of interest for our experimental runs. Charles Tripp programmed the optimizer, adapted the optimization problem to be compatible with derivative-free optimization approaches, and designed and implemented the parameterization method presented in this paper. Jennifer King, Darice Guittet, Charles Tripp, and Aaron Barker wrote, edited, and contributed to this manuscript.

*Competing interests.*   The authors declare that they have no conflict of interest.

*Acknowledgements.*   This work was authored [in part] by the National Renewable Energy Laboratory, operated by Alliance for Sustainable Energy, LLC, for the U.S. Department of Energy (DOE) under Contract No. DE-AC36-08GO28308. Funding provided by the U.S. Department of Energy Office of Energy Efficiency and Renewable Energy Wind Energy Technologies Office. The views expressed in the article do not necessarily represent the views of the DOE or the U.S. Government. The U.S. Government retains and the publisher, by accepting the
article for publication, acknowledges that the U.S. Government retains a nonexclusive, paid-up, irrevocable, worldwide license to publish or reproduce the published form of this work, or allow others to do so, for U.S. Government purposes.

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
