# Peer review of "A Simplified, Efficient Approach to Hybrid Wind and Solar Plant Site Optimization"

_Wind Energy Science, 2021_

## Referee Comment (RC1)

**Comments on manuscript WES-2021-54 "A simplified, Efficient Approach to Hybrid Wind and Solar Plant Site Optimization" by Tripp et al.**

Optimization of hybrid renewable energy systems (HRES) is a challenging topic of practical relevance with growing interest in the wind energy research community. The authors focus on site layout optimization of a utility-scale wind-solar HRES for given wind turbine specification and nameplate capacities for wind and solar. The approach is based on introducing a set of few representative parameters to be optimized via evolution strategies, where constraints are imposed via a penalty function. The approach is tested for optimizing the annual energy production (AEP) of two sites with different correlation between solar and wind resources and for two generic site boundaries. This manuscript is an interesting contribution to the literature. Nevertheless, a few questions and remarks come to mind.

**On wind turbine choice:**
Why was the 1.5MW 77m rotor diameter wind turbine chosen? Wouldn't a layout optimization be more likely applied to new plants? If so, wouldn't larger nameplates and rotor sizes, and hence fewer wind turbines for the same capacity, be expected? How would that impact the results?

**On wind to solar capacity ratio:**
What motivated the choice of wind and solar capacities? It looks like, for the specific choice here, that wind energy would be the dominant contributor to plant AEP and, hence, wake loss minimization is expected to be key. To appreciate this, it would be helpful to state the shares of wind and solar of the overall AEP.
What is the sensitivity of the results to the wind to solar capacity ratio? Would the results change if solar were the dominant AEP contribution? It may be interesting to include a case in the analysis where the choices of nameplate capacities lead to a dominant share of solar in the overall AEP.

**On the choice of optimization objective:**
As the authors state, other objectives than AEP optimization are possible and, in my view, likely in any practical application. If feasible, it would strengthen the manuscript if an optimization for net present value or internal rate of return were included.

**On the site selection:**
It may be helpful to the reader if Pearson's coefficient were explained and what range of values could be typically expected. At least stating the equation used and (re)stating the time resolution and length of the data it was computed on is needed, as monthly, annual or multi-annual complementarity could significantly vary at sites.
What was the reasoning for using Pearson's correlation instead of alternatives, e.g. Spearman's correlation? Would that lead to selecting different sites?
While the two selected sites had the lowest and highest Pearson's coefficient in the continental United States, the terms "high and low correlation location" are misleading as both values in magnitude rather indicate a lack of correlation than the implied correlated/anti-correlated behavior. Moreover, could it be that the differences in the results shown were rather driven by the difference in the wind rose than in the complementarity of wind and solar resources?

**On the interpretation of results:**
Table 2 provides large benefits over the corresponding "baseline", however, this is merely the starting point for the optimization and not a realistic alternative choice. It would be helpful to assess the method also in comparison with a reasonable baseline. For example, how would the results compare to a state-of-the-art wind farm layout optimization, where the solar panels are placed at the southern border of the site?
It looks like, for the cases shown, that the main benefits stem from wake loss minimization – is this true? Some layouts in figures 5 and 6 apparently take advantage of slightly aligning turbines out of the predicted wake. How sensitive are the results to the wake model, i.e. how reliable are the optimization results? Figure 9b could benefit from different scales and zooms.

Finally, the conclusions look more like a summary. Would be nice to get some recommendations.

---

## Author Comment (AC3)

[revised manuscript text omitted]

(a) Wind Rose    (b) Baseline    (c) Solution 3c    (d) Solution 3d    (e) Solution 3e    (f) Solution 3f

**Figure 3.** CMA-ES solutions for the low-correlation location and circular boundary.

[Figure]

(a) Baseline    (b) Solution 4b    (c) Solution 4c    (d) Solution 4d    (e) Solution 4e    (f) Solution 4f

**Figure 4.** CMA-ES solutions for the low-correlation location and irregular boundary.

[Figure]

**Figure 5.** Solutions generated for the high-correlation location and irregular boundary for a range of solar and wind capacity mixes, holding total nameplate capacity at 125MW.

entirely; however, a few competitive layouts were found that place the solar region deep in the site's interior, an interesting
20 trade-off that increases turbine spacing at the cost of some flicker and shading losses.

The solutions shown in Figure 3 are generated layouts for the low-correlation location and circular boundary. Here, we see that the more uniform and lower speed wind distribution results in very different solutions than at the high-correlation location. In response to a less concentrated wind direction distribution, the solver proposes layouts that space turbines evenly and place the solar region near the site center, giving turbines some additional separation. Similar results are shown in Figure 4 using
25 the irregular boundary, which primarily differ in an increased utilization of boundary turbines, and placement of the solar region into the northeastern corner of the site. These solutions are likely found because placing the solar in this corner actually causes boundary turbines to avoid the corner and therefore achieve increased spacing. A further-refined parameterization might specially handle border turbine placement in sharp boundary peninsulas such as this one. The ability to generate multiple competitive alternative layouts is a distinct advantage of evolution strategies and other stochastic optimization approaches.
30 Here, we see the creative power of these solution methods in finding a large diversity of viable candidate layouts, all of which yield high objective function scores. In choosing to lay out a hybrid site, one might use these methods to generate a number of good candidate sites, and then choose among them based on other important factors that are difficult to encode in such an objective function, such as ease of access, maintenance or cabling concerns, aesthetics, and more.

**0.2 Layouts for varying capacity mixes.**

35 Figure 5 shows solutions for various solar to wind generation capacity proportions while holding total capcity equal to 125 MW. For solar-heavy specifications, turbines are placed where they will never shade the solar region, and are also spread out to minimize GCR losses, with reducing wake losses only a secondary concern. Figure 5b is a supprising layout which uses the solar region to position the turbines along two rows in a way which also yields low 2.27% wake losses for this location. As solar capacity is decreased and wind capacity is increased, the solar region naturally shrinks and is gradually placed to allow
40 for reduced wake losses, with solar losses taking a back seat. Figure 5f shows a primarily wind-based layout with solar stuffed in-between two turbine rows almost as an afterthought. However, even in this case flicker losses are only 0.1% and the panels are rarely shaded.

---

## Author Comment (AC5)

**0.1 Objective Design**

[revised manuscript text omitted]

---

## Author Response (AR1)

Thank you for your careful review and insightful suggestions. Please see our responses and proposed strengthening actions below.

**Summary of changes:**

1. We clarified our rationale for using AEP at the end of Section 3.2, Objective Design
2. We clarified our rationale for using the mix of solar and wind capacities to Section 3.2.
   a. This capacity mix yields an approximate AEP balance between solar and wind at the high correlation location
3. We discussed our of choice of correlation coefficients and site selection process in the first paragraph of Section 4.
4. We explained how interior solar can sometimes be better than southern solar placements to the second paragraph of Section 3.1.
5. We added a discussion of the impacts of wind and solar capacity ratios on site layout solutions to Section 4.2 and added Figure 9, showing solutions for various solar-to-wind capacity ratios.
6. We added Table 3 which lists the performance statistics (solar and wind AEP's, losses) for each solution in Figures 5-8. This table allows the reader to more deeply explore the relative contributions of wind and solar to AEP for each layout.
7. We added discussion of land use restrictions to both the objective function section and included exploring land use issues further in the future work listed in the conclusion.

**On wind turbine choice:**

We chose the default turbine in the SAM library, knowing that it could be replaced with any size turbine desired. For this paper, the nameplate capacity of the wind and solar were fixed, and so too was the number of turbines. Using 2MW turbines we would have 37 or 38 turbines to place, and this would have made the turbine grid less dense. Without running it, it's hard to say exactly which aspects would change the most in the resulting layout candidates. However, the overall layout optimization problem and the effectiveness of this approach would be similar.

**On wind to solar capacity ratio:**

The nameplate wind capacity was fixed at 75 MW, which is roughly half of the typical pure wind power density on the circular site (approximately 5.2MW/km2 over a 3 km radius circle). Because the wind resource was stronger than the solar resource on these sites we chose a solar capacity of 50 MW. **We added a breakdown of the solar and wind contributions to AEP for the layouts shown, a discussion of the impacts of the wind to solar ratio on the layouts found, and generated layouts for a variety of wind and solar capacity ratios.**

**On the choice of optimization objective:**

We aim to provide a proof of concept that stochastic optimization of low-dimensional parametrized layouts is an effective method for producing efficient hybrid plant layouts. With that in mind, **we chose to optimize AEP because it provides a clear yet challenging objective function without the additional complexity and sensitivity to assumptions which an objective such as net present value brings with it.** NPV introduces many pricing and cost assumptions, and the resulting layouts can be highly sensitive to these assumptions, which could easily conflate and cloud the proof-of-concept we aim to demonstrate with this work. Since we fixed the nameplate capacities and we did not use a detailed cost model, the costs of each candidate don't change. Revenue in an NPV objective would change depending on how prices vary with time, whereas AEP with fixed capacities can stand-in for a constant price of energy, therefore providing an objective which is both similar to a practical objective like NPV, but simple and clear enough to provide a proof-of-concept. **We added a passage to the document clarifying our rationale for using an AEP objective.**

Additionally, we agree that interconnect utilization, particularly with land use constraints in mind, can be a useful optimization objective. To demonstrate the method's ability to optimize objectives other than AEP, **we included an additional subsection and figure showing and discussing the results of optimizing layouts with this objective for interconnect capacities ranging from 40 to 65MW, added additional discussion of interconnect utilization in the objective function section, and expanded our discussion of applying the optimizer to additional objectives to the future work section in the conclusion.**

**On site selection:**

Pearson Correlation Coefficient is most commonly used in this type of analysis (https://www.sciencedirect.com/science/article/pii/S0038092X19311831, https://www.nrel.gov/docs/fy13osti/57816.pdf, https://ieeexplore.ieee.org/stamp/stamp.jsp?tp=&arnumber=8749030). It is true that Pearson Correlation Coefficient relies on the assumption of bi-variate normality for the two variables while no distributional assumptions are required for Spearman's Rank Correlation Coefficient. However, Spearman's CC is generally considered more appropriate for use on ordinal (i.e. ranked) data. Furthermore, Pearson is most commonly used for timeseries data.  When we examined the dataset closely using multiple correlation coefficients (including both Pearson and Spearman), Pearson CC was most reflective of the complementarity seen. **We added discussion of our choice of correlation coefficients and additional supporting details on the example site selection process to the document,** Including the equation and nature of the data used in selecting the test sites. Both wind rose and resource correlations impacted our AEP objective; it is expected that different site layouts would be found for different resource correlations or wind roses. We amended our analysis of the results to reflect this fact.

**On the interpretation of results:**

**You are correct: placing solar on the interior of the site can allow for greater separation between turbines and therefore reduce wake losses.** When this is done in a way which reduces wake losses by more than the flicker and shading losses which might be incurred by an interior solar placement, the interior solar placement layout is superior to a southern boundary placement. *In this sense, the largest benefits from allowing flexible solar placement are typically in reducing wake losses.* **We updated our text to clarify this point, and to motivate the flexible placement of the solar region.**

The prior distribution on the solar region placement ("solar y position") used to initialize the optimization is set in a way which biases the search towards exploring candidates which place the solar region along the southern boundary of the site. In Figure 10's optimizer trajectory plot we see that the optimizer indeed often finds good layouts which have solar regions on or near the southern boundary of the site. However, as seen in Figures 5-8, these are not the only high-performance layouts for any of the four example scenarios.

Additional concerns and tradeoffs must be made even if layouts are constrained to solar along the southern boundary, such as where along the boundary to place the solar region ("solar x position"), what shape and density of solar region to use as lower solar densities reduce internal shading but also increase wake losses by increasing turbine density ("solar gcr" and "solar aspect power"), and how large of a setback to use between the solar region and turbines, which trades off turbine-solar shading and flicker losses for wake losses ("solar x buffer" and "solar s buffer").

Stanley et al.'s "Massive simplification of the wind farm layout optimization problem" demonstrated that the turbine parameterization we used performs competitively with non-parameterized state of the art layout optimization. Therefore, if we constrained solar placement to a fixed reasonable placement along the southern boundary, and only optimized the turbine layout, we will find solutions which are comparable with a state-of-the-art turbine layout optimization, where the solar region is placed at a fixed region along the southern border of the site.

As with many turbine layout optimization methods, the results are moderately sensitive to the wake model. However, the restrictions imposed by parameterizing turbine placement mitigate wake model sensitivity in comparison to a non-parameterized approach. A non-parameterized optimizer's flexibility allows it to make micro-siting adjustments, moving individual turbines just right to exploit weaknesses in the wake model. **In contrast, our parameterized turbine grid simply cannot make these micro-adjustments and therefore is less likely to generate layouts which depend on artifacts of the particular wake model used and which may have little real-world benefit.** One way of thinking about this effect is that the parametrization applies a strong regularization to the optimization problem, which increases the robustness of solutions and decreases solution sensitivity to the peculiarities of any models used.

**On Land use restrictions:**

We agree that land use restrictions are an important concern for many hybrid plant installations. Land use constraints can be applied to the optimization process in the same way that boundary constraints were applied: solar modules were simply excluded from disallowed zones while turbines were moved to the nearest valid location (if no valid location was identified, they were simply removed from the layout). By associating a quadratic penalty with invalid turbine placements, we encourage the solver to focus its efforts on generating valid layouts. Variable land use costs could also be incorporated in a similar fashion if these costs are accounted for by the objective function. **We added discussion of land**

use restrictions to both the objective function section and included exploring land use issues further in the future work listed in the conclusion.

---

## Referee Report (RR1)

WES-2021-54

Overall
- Nice work and very interesting. As written, the paper undercuts the novelty of the work in moving towards physical design optimization of HPP (see detailed notes below). The core contributions surround the learnings related to HPP design under different conditions. Too much emphasis is placed on the mechanics of the optimization rather than the results (which are really interesting!). Again, see deatailed notes below.
- Lots of aronyms are used without definiton on first use – it is particularly important to correct this as many of them are solar related and this is a wind journal. People will not know these
- Generally, the article could benefit from a primer for wind people on solar. It is particularly hard to follow section 2.3 which seems to be a very interesting and relevant contribution of the work

Abstract
- Avoid acronyms in the abstract – if used, you need to put them next to the word on first use (i.e. line 7)
- What is scientifically interesting about the work? The tutorial is not really a scientific contribution. Consider replacing the last sentence with something of interest that was discovered in the optimization process – surprising trends in the designs, trade-offs that were significant, etc

Introduction
- It would be helpful to define hybrid power plants in the intro – dont assume the reader is familiar or has the same understanding of HPPs
- Many WFLO problems in literature focus on cost of energy or cost/energy – work looking only at energy optimization is a bit outdated
- I think it is important to distinguish a bit more on the topics of hybrid power plant optimization problems. You mention sizing – there is a TON of literature in this space and most of these fall into the category of MILP sine they focus on sizing the assets time-series energy production. Here you are going BEYOND sizing to look at physical design – which is a a nascent area wehre little research has been done. Make sure that message is clear in the abstract, intro and conclusions
- Please remove the section 1.1 and transform this into a paragraph. Bullets should onlybe used if absolutely necessary and they are not here.
- The last sentence in section 1 "We aim…" reads a bit funny… maybe just say, In this work, we provide a proof of concept of stochastic optimization of low-d parameterized layouts as an effective method…
- Consider adding a paper roadmap at the very end of section 1

Hybrid plant model

- For sections 2.1 and 2.2 can you elaborate a bit more on the limitations of the selected wind and solar plant model – there are many model choices here and they arent well justified
- Section 2.3 is hard to follow. Figure 1 is particularly interesting but only 3 time steps per hour seems like pretty low resolutionn- is there any validation of this?
- Figure 2 bounds dont need to be so big as there is negligible effects beyond +-200 and are these meters? Nothing is labeled
- Consider putting a picture ahead of figures 1 and 2 that shows the layout of the turbines and PV being simulated. Without having such a visual, its hard to tell what is going on… there is a lot of information described in text where diagrams would be helpful
- Is this model described anywhere else? I dont see any citations. If there is not enough space in situ, an appendix that more thoroughly describes the model would be helpful

Optimization methodology
- Again, some diagrams could be helpful here – using the baseline plants for example. It is hard to follow table 1 on first inspection. I had to reread the section several times and scrolled down to figure 5 and 6 to in any case to help interpret it
- I think it is fine to choose AEP as this a first study of this type so it is good to start there rather than add additional complexity. The long discussion is not necessary and could be moved to future work. Again, its important to emphasize in the introduction that this is a physical design study to differentiate from all the work on sizing of HPPs that already exists
- It seems there is a lot of work going into the constraints handling that is manually programmed. Can you describe this more in an appendix or refer to code documentation? Generally it would be nice to see references to the code here
- ES is a good starting point but certainly an area for future work as well
- Do you have a reference on random search?

Experimental results
- Consider using a table for the properties of the two sites. Again, a lot of things are described in text where diagrams or tables would be better
- Inte3esting that the high correlation sites have a lot more spread in terms of AEP gains… i'd like to see more discussion on this and explanation
- In section 4.1, A core scientific contribution of this paper is on how the difference in correlation supports different trends (exploiting trade-offs differently) in system design. I would have liked to see a partitioning of the effects of the correlation versus the wind rose. It would be nice to see the wind roses swapped to tease apart the effect of the strength of directionality of the wind rose versus the strength of the correlation in terms of influencing the design trends. Maybe you can speak to this a bit more without having to do the optimizations themselves…
- Section 4.4 can be an appendix – instead it would be nie to see more elaboration on sections 4.2 and 4.3 – the value of the paper is in explaining and understanding the influence of site conditions and problem formulation on design trends for HPPs. The particulars of the algorithm are secondary

Conclusions
- I recommend rewriting the conclusions completely. The emphasis should be on the results and interpretation of the HPP design optimization – not the optimization mechanics.
- Future work could be extended quite a bit – a lot of the discussion in 3.2 could be brought here

---

## Author Response (AR2)

Thank you for your thoughtful review of our manuscript, 'A Simplified, Efficient Approach to Hybrid Wind and Solar Plant Site Optimization'. We have made changes in response to each of your comments, and with your help we hope to have produced a significantly strengthened manuscript.

**Overall**

**- Nice work and very interesting. As written, the paper undercuts the novelty of the work in moving towards physical design optimization of HPP (see detailed notes below). The core contributions surround the learnings related to HPP design under different conditions. Too much emphasis is placed on the mechanics of the optimization rather than the results (which are really interesting!). Again, see detailed notes below.**

Thank you for your perspective. We have adjusted the abstract, introduction, and conclusion to place greater emphasis on the results and interpretation. Additionally, we have made changes throughout the manuscript to place less focus on the optimization mechanics and more on the results. In particular, additional interpretation has been added to each experimental result subsection in Section 4.

**- Lots of acronyms are used without definition on first use – it is particularly important to correct this as many of them are solar related and this is a wind journal. People will not know these.**

We have reviewed every acronym used and ensured that it is defined when first used.

**- Generally, the article could benefit from a primer for wind people on solar. It is particularly hard to follow section 2.3 which seems to be a very interesting and relevant contribution of the work**

We have added a PV design and PV modeling primer to Section 2.

**Abstract**

**- Avoid acronyms in the abstract – if used, you need to put them next to the word on first use (i.e. line 7)**

We removed all acronyms from the abstract.

**- What is scientifically interesting about the work? The tutorial is not really a scientific contribution. Consider replacing the last sentence with something of interest that was discovered in the optimization process – surprising trends in the designs, tradeoffs that were significant, etc Introduction**

We revised the last sentence to emphasize our findings, as well as reworking the last paragraph of the introduction to focus on these findings rather than the tutorial, which has been moved to Appendix A.

**Introduction**

**- It would be helpful to define hybrid power plants in the intro – don't assume the reader is familiar or has the same understanding of HPPs**

We added a definition of HPPs in the first sentences of the intro, and additional supporting information as needed.

**- Many WFLO problems in literature focus on cost of energy or cost/energy – work looking only at energy optimization is a bit outdated**

We added a mention of LCOE optimization in WFLO, along with a few recent citations using LCOE as an objective. We added a mention of using the Financial Models for calculating NPV to Section 2.

**- I think it is important to distinguish a bit more on the topics of hybrid power plant optimization problems. You mention sizing – there is a TON of literature in this space and most of these fall into the category of MILP since they focus on sizing the assets time-series energy production. Here you are going BEYOND sizing to look at physical design – which is a a nascent area where little research has been done. Make sure that message is clear in the abstract, intro and conclusions**

We added a sentence discussing this to the first paragraph of the introduction, a clear statement to this effect to the abstract, and a statement to this effect in the beginning of the conclusion.

**- Please remove the section 1.1 and transform this into a paragraph. Bullets should only be used if absolutely necessary and they are not here.**

We removed Section 1.1 and reworked it's content into the second to last paragraph of the introduction.

**- The last sentence in section 1 "We aim…" reads a bit funny… maybe just say, In this work, we provide a proof of concept of stochastic optimization of low-d parameterized layouts as an effective method…**

We rewrote this sentence as suggested and incorporated it into the reworked final two paragraphs of the introduction.

**- Consider adding a paper roadmap at the very end of section 1 Hybrid plant model**

This suggestion motivated us to rework the last two paragraphs of the introduction and to combine the paper roadmap with the reworked contribution list, which can now be found in the second to last paragraph of the conclusion.

 Hybrid Plant Model

**- For sections 2.1 and 2.2 can you elaborate a bit more on the limitations of the selected wind and solar plant model – there are many model choices here and they aren't well justified**

We added a passage to Section 2.1 justifying our wake model choice and it's limitations, and another to Section 2.2 discussing our PV model parameters and choices.

**- Section 2.3 is hard to follow. Figure 1 is particularly interesting but only 3 time steps per hour seems like pretty low resolution- is there any validation of this?**

We added a discussion of Figure 1 (now Figure 2), including that it was generated as a demonstration using a short time window and lower resolution than used in our model.

**- Figure 2 bounds don't need to be so big as there is negligible effects beyond +-200 and are these meters? Nothing is labeled**

We added meters to labels and updated the caption to address this concern.

**- Consider putting a picture ahead of figures 1 and 2 that shows the layout of the turbines and PV being simulated. Without having such a visual, its hard to tell what is going on… there is a lot of information described in text where diagrams would be helpful**

We added a sub figure to Figure 1 (now Figure 2) showing how the PV modules are laid out all around the turbines in a sub-grid within the turbine layout. Additionally, we added a figure to the beginning of Section 2 visualizing an example layout and setting the stage visually for the reader to contemplate the hybrid plant layouts discussed later in the text.

**- Is this model described anywhere else? I don't see any citations. If there is not enough space in situ, an appendix that more thoroughly describes the model would be helpful Optimization methodology**

We added a description of the shadow flicker model to Section 2.3, and a link to the source code implementation of the model.

**Optimization Methodology**

**- Again, some diagrams could be helpful here – using the baseline plants for example. It is hard to follow table 1 on first inspection. I had to reread the section several times and scrolled down to figure 5 and 6 to in any case to help interpret it**

We found it quite difficult to produce a meaningful diagram, but have clarified the descriptions of the parameters, added Figure 1 visualizing a typical layout under this scheme before the parameters are presented, made the reference to further discussion of the turbine layout parameters more prominent, and added links to the implementation of the parametric layout mapping parameter values to physical layouts.

**- I think it is fine to choose AEP as this a first study of this type so it is good to start there rather than add additional complexity. The long discussion is not necessary and could be moved to future work. Again, its important to emphasize in the introduction that this is a physical design study to differentiate from all the work on sizing of HPPs that already exists**

We simplified the discussion of objective choice and moved it into the future work section of the conclusion. We adjusted the language here and in the abstract, intro, and conclusion to emphasize physical optimization as primary contribution of this work.

**- It seems there is a lot of work going into the constraints handling that is manually programmed. Can you describe this more in an appendix or refer to code documentation? Generally, it would be nice to see references to the code here**

We added a link to the source code repository to contributions section. We added a URL pointing to the exact block of code implementing the mapping of parameter values to physical locations of turbines and solar modules to Section 3, and we added a third link to the exact block of code applying soft constraints to the optimization problem.

**- ES is a good starting point but certainly an area for future work as well**

Agreed. We added a passage to this effect to the future work section.

**- Do you have a reference on random search? Experimental results**

Random search is such a basic strategy that it is not generally considered as an official approach. Fundamentally RS is just blindly generating random layouts from a fixed distribution. We added a passage to the RS section clarifying its use here as a simple baseline for evaluating the benefits of more complex ES algorithms.

Experimental Results

**- Consider using a table for the properties of the two sites. Again, a lot of things are described in text where diagrams or tables would be better**

We added a table to the beginning of this section providing a clear comparison of the two sites.

**- Interesting that the high correlation sites have a lot more spread in terms of AEP gains… I'd like to see more discussion on this and explanation**

We agree. Additional discussion into this trend shown in the data adds significant value to the manuscript. In response, we have added several sentences discussing these results, proposing a possible explanation, and potential resulting design guidelines.

**- In section 4.1, A core scientific contribution of this paper is on how the difference in correlation supports different trends (exploiting trade-offs differently) in system design. I would have liked to see a partitioning of the effects of the correlation versus the wind rose. It would be nice to see the wind roses swapped to tease apart the effect of the strength of directionality of the wind rose versus the strength of the correlation in terms of influencing the design trends. Maybe you can speak to this a bit more without having to do the optimizations themselves…**

We agree. Just as with the previous point, we expanded the discussion of the influence of resource correlation on optimized layouts, and pointed to future work to further elucidate the impact of resource correlation and other factors of interest on design guidelines.

**- Section 4.4 can be an appendix – instead it would be nice to see more elaboration on sections 4.2 and 4.3 – the value of the paper is in explaining and understanding the influence of site conditions and problem formulation on design trends for HPPs. The particulars of the algorithm are secondary.**

We agree. We moved Section 4.4 into Appendix A. Just as with the previous two points, we significantly expanded sections 4.2 and 4.3. In Section 4.2, we added discussion and interpretation of the results and the possible design influences of various mixes of solar and wind generation. We proposed explanations and design guidelines informed by our optimization results. In Section 4.3, we added further

interpretation of the results using varying interconnect capacities and a discussion of the design principles these results hint at.

**Conclusions**

**- I recommend rewriting the conclusions completely. The emphasis should be on the results and interpretation of the HPP design optimization – not the optimization mechanics.**

We have completely rewritten the conclusion, placing a strong emphasis on the results and their interpretation, and the possible design considerations these results imply.

**- Future work could be extended quite a bit – a lot of the discussion in 3.2 could be brought here**

We expanded the future work discussion to cover many of the points from Section 3.2, as well as motivating further investigation into the trends and possible design guidelines revealed in Section 4.

Thank you again for your time and valuable feedback. Your assistance has helped us meaningfully improve our manuscript.

Best,

Charles, Darice, Jen, and Aaron.